# Substrates bind to residues lining the ring of asymmetrically engaged bacterial proteasome activator Bpa

Tatjana von Rosen[1,5], Rafal Zdanowicz [1,4,5], Yasser El Hadeg [1,5], Pavel Afanasyev [2], Daniel Boehringer [2], Alexander Leitner [3], Rudi Glockshuber [1] & Eilika Weber-Ban [1] ✉

Mycobacteria harbor a proteasome that was acquired by Actinobacteria through horizontal gene transfer and that supports the persistence of the human pathogen *Mycobacterium tuberculosis* within host macrophages. The core particle of the proteasome (20S CP) associates with ring-shaped activator complexes to degrade protein substrates. One of these is the bacterial proteasome activator Bpa that stimulates the ATP-independent proteasomal degradation of the heat shock repressor HspR. In this study, we determine the cryogenic electron microscopy 3D reconstruction of the complex between Bpa and its natural substrate HspR at 4.1 Å global resolution. The resulting maps allow us to identify regions of Bpa that interact with HspR. Using structure-guided site-directed mutagenesis and in vitro biochemical assays, we confirm the importance of the identified residues for Bpa-mediated substrate recruitment and subsequent proteasomal degradation. Additionally, we show that the dodecameric Bpa ring associates asymmetrically with the heptameric α-rings of the 20S CP, adopting a conformation resembling a hinged lid, while still engaging all seven docking sites on the proteasome.

Bacteria rely on controlled protein degradation for cellular quality control and regulation. In addition to canonical bacterial chaperone-proteases, Actinobacteria have acquired a eukaryotic-like proteasomal degradation system[1]. Components of this system are not required under standard laboratory culture conditions but are essential for Actinobacteria to adapt to rapid changes in their environment[2]. For example, the proteasome is crucial for the human pathogen *Mycobacterium tuberculosis* to persist inside host macrophages[3,4].

The bacterial proteasome core particle (20S CP), like its eukaryotic counterpart, is a cylinder comprised of four stacked heptameric rings[5–7]. The two inner β-rings carry the proteolytic active sites and the two outer α-rings gate the access to the proteolytic chamber[5]. The proteasome associates with ring-shaped activators that unfold and

translocate substrates into the proteolytic chamber. An ATPase ring complex called Mpa in mycobacteria and ARC in all other Actinobacteria engages the proteasome by inserting its C-terminal GQYL motif, similar to the HbYX motif of the regulatory 19S particle ATPases in eukaryotes, into binding pockets located between the α-subunits of the proteasome[8–10]. The interaction between Mpa/ARC and the 20S CP induces a conformational change in the α-subunits that leads to opening of the gate for access to the proteolytic chamber.

This proteasome/unfoldase system degrades proteins that have been post-translationally modified with the prokaryotic ubiquitin-like protein (Pup) through a process called "pupylation"[9,11]. Pupylation, unlike ubiquitination in eukaryotes, is carried out by a single ligase, proteasome accessory factor A (PafA), that catalyzes the covalent

[1]Institute of Molecular Biology and Biophysics, ETH Zurich, Zurich, Switzerland. [2]Cryo-EM Knowledge Hub (CEMK), ETH Zurich, Zurich, Switzerland. [3]Institute for Molecular Systems Biology, ETH Zurich, Zurich, Switzerland. [4]Present address: International Institute of Molecular Mechanisms and Machines, Polish Academy of Sciences, Warsaw, Poland. [5]These authors contributed equally: Tatjana von Rosen, Rafal Zdanowicz, Yasser El Hadeg. ✉e-mail: eilika@mol.biol.ethz.ch

attachment of Pup to lysine side chains of a diverse range of substrates[12–14]. By association of Pup with the coiled-coil domains of Mpa, substrates are recruited to the Mpa-20S CP complex[15,16]. The N-terminal region of Pup is then engaged into the Mpa pore and the ATPase-driven motion of Mpa pore loops unfolds the substrates and translocates them into the proteolytic chamber for degradation[9,17,18].

Alternatively, the 20S CP can associate with the bacterial proteasome activator (Bpa) and degrade proteins in a pupylation-independent manner[19,20]. Bpa monomers feature a four-helix bundle fold and assemble into a dodecameric, ring-shaped complex with the protomers arranged around a large central opening of about 40 Å in diameter[21,22]. Bpa bears no structural or sequence homology to Mpa, but engages the 20S CP via the identical C-terminal GQYL motif[21,22]. Residues Y173 and L174 of the Bpa GQYL motif make contacts with R26 and K52 of the proteasome α-subunits, respectively, stimulating an outward rotation of the gating H0 helices of the α-subunits[21,23]. Unlike Mpa, Bpa facilitates the proteasomal degradation of its substrates without making use of a tagging system like pupylation and without the expense of ATP.

In contrast to eukaryotes, where PA28/PA26 (also known as 11S) and PA200 complexes exist[24–27], Bpa is the only known ATP-independent proteasome activator in bacteria. By investigating the Bpa-dependent degradation of model substrates and of the natural substrate HspR, we previously demonstrated that disorder is required, but not sufficient for Bpa-mediated proteasomal degradation in mycobacteria[28]. HspR regulates the transcription of important heat-shock chaperones and therefore plays a role in survival of bacteria under stress[29–33]. Upon heat shock, HspR partially unfolds and detaches from the promoter sequence[33]. In the unbound state, HspR is recruited to Bpa-mediated proteasomal degradation through residues located in its N-terminal DNA-binding domain[28]. Subsequently, the disordered C-terminal stretch of HspR acts as a threading motif to promote efficient degradation[28]. While the binding mode on the side of HspR is well understood, the mechanism of substrate recognition at the activator remains elusive.

In our study, we investigate the interaction between Bpa and its substrate using cryogenic electron microscopy (cryo-EM). We determine the structure of Bpa from *M. tuberculosis* in complex with HspR and discover that the substrate binds to the lower portion of the Bpa ring. Guided by the structural information, we use site-directed mutagenesis and in vitro degradation assays to identify specific residues on the Bpa surface involved in substrate binding and in facilitating proteasomal degradation of HspR. Our findings provide insight into the ATP-independent proteasomal degradation mechanism in the human pathogen *M. tuberculosis*.

## Results

### Bpa associates co-axially with the proteasome α-rings and is tethered asymmetrically while engaging all proteasomal binding sites

We had previously employed genetically encoded, site-specific cross-linking to stabilize the complex between Bpa and a proteasome variant lacking the first seven residues of the α-subunits (open gate proteasome), allowing us to obtain cryo-EM 3D reconstructions of Bpa-CP complexes, where tethering of Bpa to one side of the proteasome and apparent upward tilting of Bpa was observed[21]. In a different study, a structure of full-length proteasome in complex with a Bpa variant lacking the flexible linker between helix H4 and the C-terminal GQYL motif (Bpa$_{\Delta155–166}$) was determined showing no tilting of Bpa relative to the proteasome[23]. However, no structure of full-length Bpa in complex with full-length proteasome (closed gate) has been solved to date.

To stabilize the full-length Bpa-CP complex for structural studies, we employed gradient fixation (GraFix)[34,35]. By combining a linear sucrose gradient with a linear gradient of glutaraldehyde crosslinker, we obtained complexes which exhibited higher molecular weight than

Bpa and 20S CP when visualized by SDS-PAGE, indicative of cross-linked species (Fig. 1A). With the additional aim of visualizing substrate on the Bpa-CP complex, we used a degradation-inactive proteasome variant (20S CP$^{T1A}$) and added the natural substrate HspR to the fractions containing the crosslinked species (Bpa-CP/HspR) (Fig. 1A, boxed in red). Grids for cryo-EM were prepared from this sample and the recorded micrographs revealed the presence of evenly distributed, doubly capped, co-axial Bpa-CP complexes (Fig. 1B). Our cryo-EM analysis resulted in 2D class averages that confirmed high occupancy of the 20S CP with Bpa and showed essentially no unbound 20S CP particles (Fig. 1C). Furthermore, the majority (more than 90%) of the core particles were capped by Bpa on both ends, a population less dominant in prior studies[21–23]. We determined the cryo-EM structure of the full-length Bpa-CP complex at a global resolution of 3.3 Å (Fig. 1D, Supplementary Fig. 1, 2, and 3, Supplementary Table 1). While no substrate density could be resolved within the low-resolution density for Bpa, by focusing our analysis on the α-rings of the 20S CP, Bpa, and the interface between these two components, we could identify different tilting angles of the dodecameric Bpa ring relative to the seven-fold symmetric proteasome (Fig. 1E). Furthermore, our analysis revealed that seven consecutive Bpa protomers docked via their C-terminal extensions into the seven GQYL-binding pockets available on the α-ring of the 20S CP, leaving the five remaining Bpa protomers unbound (Fig. 1F). This mode of binding resulted in strong tethering of one side of Bpa to the proteasome, leading to an asymmetrical arrangement resembling a "hinged lid". The map solved without applying seven-fold symmetry revealed that all proteasomal binding pockets were occupied, however the Bpa densities within the pockets exhibited weaker signal compared to the 20S CP, likely due to some level of conformational heterogeneity of the residues preceding the GQYL motif.

To probe the asymmetric interaction, we generated a covalent dimer consisting of a C-terminally truncated Bpa lacking the C-terminal 18 residues and a full-length Bpa (BpaΔC-Bpa). This construct was used in a previous study, where it was shown that the dimers assemble into a 12-membered Bpa ring that contains six evenly distributed C-terminal interaction motifs[23]. For negative stain EM analysis, we assembled complexes of the inactive full-length 20S CP with either the engineered Bpa ring, composed of BpaΔC-Bpa dimers, or the wild-type Bpa ring, following the GraFix protocol. Using the collected datasets, we performed 2D classification and low-resolution 3D reconstructions (Supplementary Fig. 4). The results reveal that the BpaΔC-Bpa dimer ring exhibits much lower affinity for the CP, with approximately 73% of proteasomes remaining uncapped and around 27% being singly capped by the BpaΔC-Bpa ring (Supplementary Fig. 4A, left). In contrast, more than 80% of proteasomes were doubly capped when the wild-type Bpa ring was used (Supplementary Fig. 4A, middle). This likely arises at least partially from an avidity effect due to the reduced total number of tails in the BpaΔC-Bpa ring. Additionally, the fact that the six tails are not consecutive may also be a contributing factor. Interestingly, the 3D reconstruction of the proteasome complex with the BpaΔC-Bpa ring revealed that it closely resembles the wild-type Bpa–proteasome complex (Supplementary Fig. 4B), since it also exhibits asymmetric tethering and adopts a "hinged lid" conformation. However, not all six available tails are involved in the interaction, as indicated by the narrower density in the EM map between the Bpa ring and the CP. This could suggest that also in the case of the mixed oligomer only the tails from one side of the Bpa ring are engaged.

To explore the interface between the Bpa activator and the 20S CP α-rings, we performed signal subtraction to focus only on the 20S CP, allowing us to solve its structure at a resolution of 2.5 Å, with D7 symmetry imposed (Fig. 2A, B, Supplementary Figs. 1, 2, and 3, Supplementary Table 1). In proximity to the H0 helices of the proteasomal α-subunits, we discerned distinct densities corresponding to the Bpa

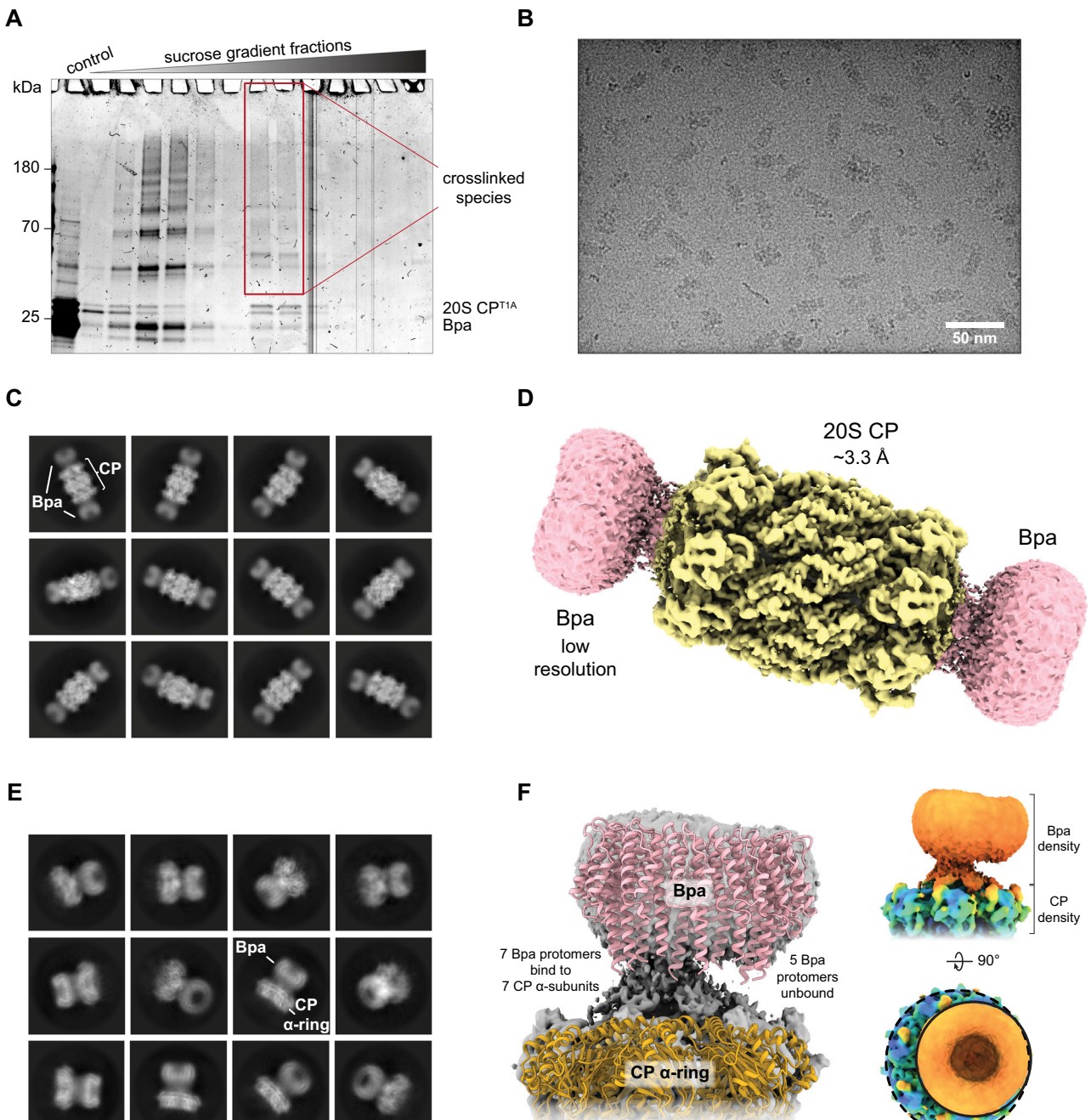

**Fig. 1 | Gradient fixation and cryo-EM analysis of the Bpa-CP/HspR sample.**
**A** Coomassie-stained SDS-PAGE of the Bpa-CP sample after gradient fixation (GraFix). Higher molecular weight species, indicative of crosslinked complexes, were observed above the bands of Bpa monomers and the monomeric 20S CP subunits. Crosslinked species in fractions isolated for structural experiments are boxed in red. The control in the first lane refers to the uncrosslinked sample before GraFix. **B** Representative micrograph from the cryo-EM data collection. **C** 2D class averages generated from the Bpa-CP/HspR dataset. 20S CP is capped at both ends by full-length Bpa. **D** 3D reconstruction of the doubly capped Bpa-CP complex. **E** 2D class averages of the Bpa-CP /HspR sample subjected to signal subtraction leaving one Bpa cap and the 20S CP α-ring. The multiple orientations that Bpa can adopt relative to the proteasome are shown. **F** Crystal structures of Bpa

(PDB 5LFJ) and the 20S CP (PDB 5LZP) were rigid-body fitted to the cryo-EM map (left). The association between the Bpa dodecamer (pink) and the 20S CP α-subunit heptamer (yellow) involves seven neighboring Bpa protomers, leaving five protomers unbound. This binding mode allows for considerable tilting of the Bpa dodecamer relative to the 20S CP α-subunit heptamer and contributes to particle heterogeneity. The cryo-EM map was modified using OccuPy[65] to amplify the density of Bpa. In addition, the same map was low-pass-filtered to attenuate high-frequency signal and was colored according to the local scale estimated by OccuPy (right). Two views of the same map show Bpa (orange density) tethered asymmetrically to the α-ring of 20S CP (green and blue density) with all 20S CP binding sites occupied by Bpa.

GQYL interaction motif (Fig. 2C). The last six residues of the Bpa C-terminal tails (169-174) could be confidently built into the density. This region exhibited improved resolution compared to our previously published cryo-EM map of the 20S CP in complex with Bpa, where side chains for only the last three residues could be modeled[21]. At lower

map thresholds, the continuous density could be traced further, reaching approximately up to residue K162. Interestingly, the orientation of this density preceding the GQYL motif indicated that the C-terminal extensions of Bpa approach the binding pockets parallel to the two proximal H0 helices of the 20S CP, rather than directly from

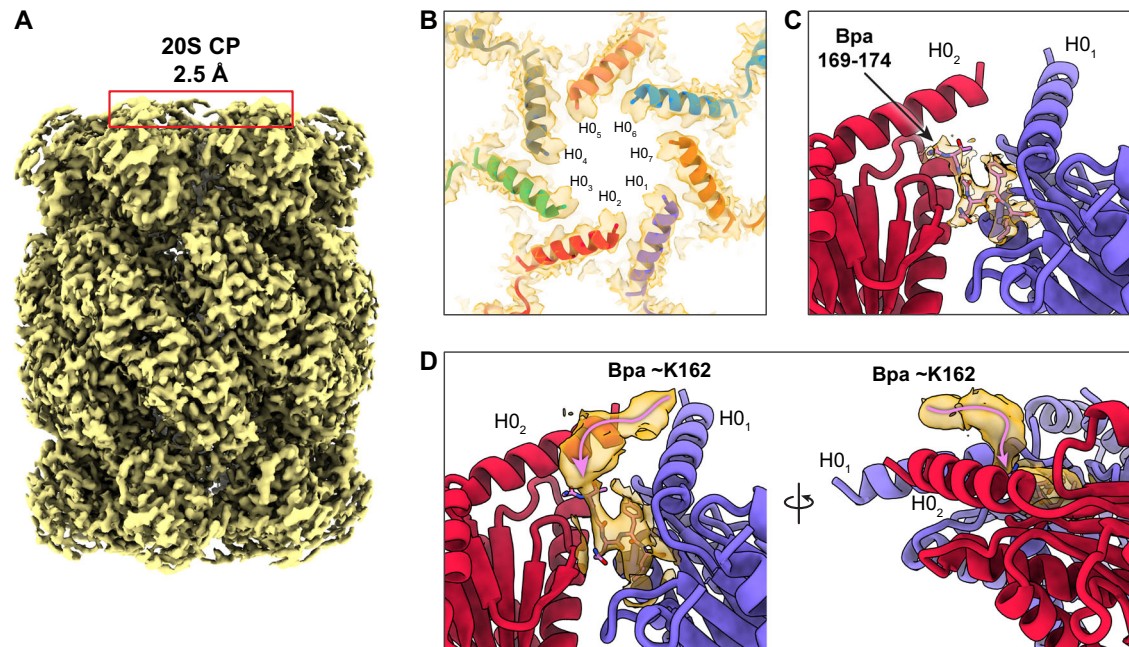

**Fig. 2 | Binding of Bpa to the 20S CP. A** Cryo-EM map of the 20S CP, from which the Bpa signal has been subtracted. The 20S CP map was reconstructed in D7 symmetry to a resolution of 2.5 Å. **B** Top view of the proteasomal gate, showing H0 helices of the α-subunits (red box in **A**). **C** A close-up view of the 20S CP binding pocket for the Bpa C-terminal GQYL interaction motif. Residues 169-174 of Bpa interacting with the 20S CP by insertion between two α-subunits of 20S CP are shown. **D** Additional densities attributed to Bpa (approx. residues 162-174) interacting with the 20S CP α-subunits. Although the poor density hindered unambiguous assignment of the residues preceding G169, the backbone of the Bpa C-terminal tail oriented parallel to the two neighboring helices can be traced (pink line).

above (Fig. 2D). Such behavior suggests that additional contacts between the Bpa C-terminal region and the proteasomal H0 helices further stabilize Bpa on the proteasome. This could explain the concentric appearance of the connecting density between one side of Bpa and the 20S CP (Fig. 1F), where the interacting side of Bpa sits nearly central above the α-ring pore whereas the Bpa ring as a whole is offset from the center.

To confirm the formation of contacts between Bpa C-terminal extensions and CP H0 helices, we assembled a complex between wild-type Bpa ring and a variant of the inactive full-length CP, and stabilized it via GraFix. In this CP variant, we mutated residues in the H0 helix of the α-subunits, which, according to our cryo-EM map, are oriented toward the Bpa density (E10A-R14A). The complex was then analyzed using negative stain EM and 2D class averages were generated (Supplementary Fig. 4A, right). Our results show that while the H0 mutant can associate with Bpa, only 31% of proteasomes are doubly capped by Bpa, with more than 60% being singly capped. By comparison, more than 80% of proteasomes with wild-type H0 helices had two Bpa rings bound, as described above. Moreover, 5.6% of H0 mutant proteasomes are not associated with Bpa at all, whereas less than 1% of such population was observed for proteasomes carrying wild-type H0 helices.

Taken together, our findings suggest that seven consecutive Bpa protomers engage and occupy all binding pockets on the proteasome α-rings, thereby tethering one side of Bpa asymmetrically to the proteasome. The insertion of the Bpa GQYL motifs into the binding pockets of the 20S CP and the resulting gate-opening is further stabilized by residues preceding the motif, which are likely forming contacts along the proteasomal H0 helices of the α-subunits.

### Cryo-EM structures of free full-length and substrate-bound Bpa from *M. tuberculosis*

To confirm that full-length Bpa in its unbound state behaves similarly to the previously studied truncated Bpa constructs, we determined its structure using cryo-EM. While high-resolution structures of truncated Bpa have been determined previously using X-ray crystallography[21,22], the conformation of the full-length Bpa dodecamer in its unbound state has not been addressed. The cryo-EM map of full-length Bpa was resolved at 3.2 Å with local resolution ranging from 2.7 to 3.4 Å (Fig. 3A–C, Supplementary Fig. 1, 2, and 3, Supplementary Table 1). Similar to the crystal structures published previously, our model confirmed that the Bpa protomer forms a tight four-helix bundle (Fig. 3D). However, in contrast to existing models, our structure showed that the helical extensions of helix H4 exhibit greater flexibility (Fig. 3E). It appears that the crystal packing in the X-ray crystallography structure of truncated Bpa (residues 15–153), resulted in stabilization of helix H4 up to residue L148[21]. Our cryo-EM maps showed no discernable density beyond Bpa residue Q140 within the H4 extension, indicating that the residues following Q140, while possessing helical propensity, are dynamic. At the N-terminal side, the first Bpa residue that could be resolved confidently was S36 which is located toward the same ring face of Bpa as the C-terminal H4 extension. This demonstrates that the first 35 residues are disordered and emanate from the Bpa face oriented toward the CP α-ring in the Bpa-CP complex.

After confirming that full-length Bpa features unstructured N- and C-terminal regions, we proceeded to study the interaction between Bpa and HspR. To minimize potential interference from these flexible regions, which could hinder accurate particle alignment during 3D reconstruction, and to clearly distinguish structural features specific to the substrate, we opted to use a truncated Bpa variant lacking the flexible N- and C-terminal segments (Bpa$_{36-139}$). Additionally, to overcome the low solubility of full-length HspR, we used a truncated HspR variant that lacked the C-terminal nine residues (HspR$_{\Delta C9}$) to generate the Bpa-substrate complex. We previously demonstrated that this truncated variant retains the ability to bind Bpa[28].

To stabilize the Bpa$_{36–139}$-HspR$_{\Delta C9}$ interaction during cryo-EM grid preparation, we subjected the complex to batch crosslinking with glutaraldehyde (GA). The crosslinked sample was then fractionated by gel filtration, followed by mass spectrometric analysis and assessment

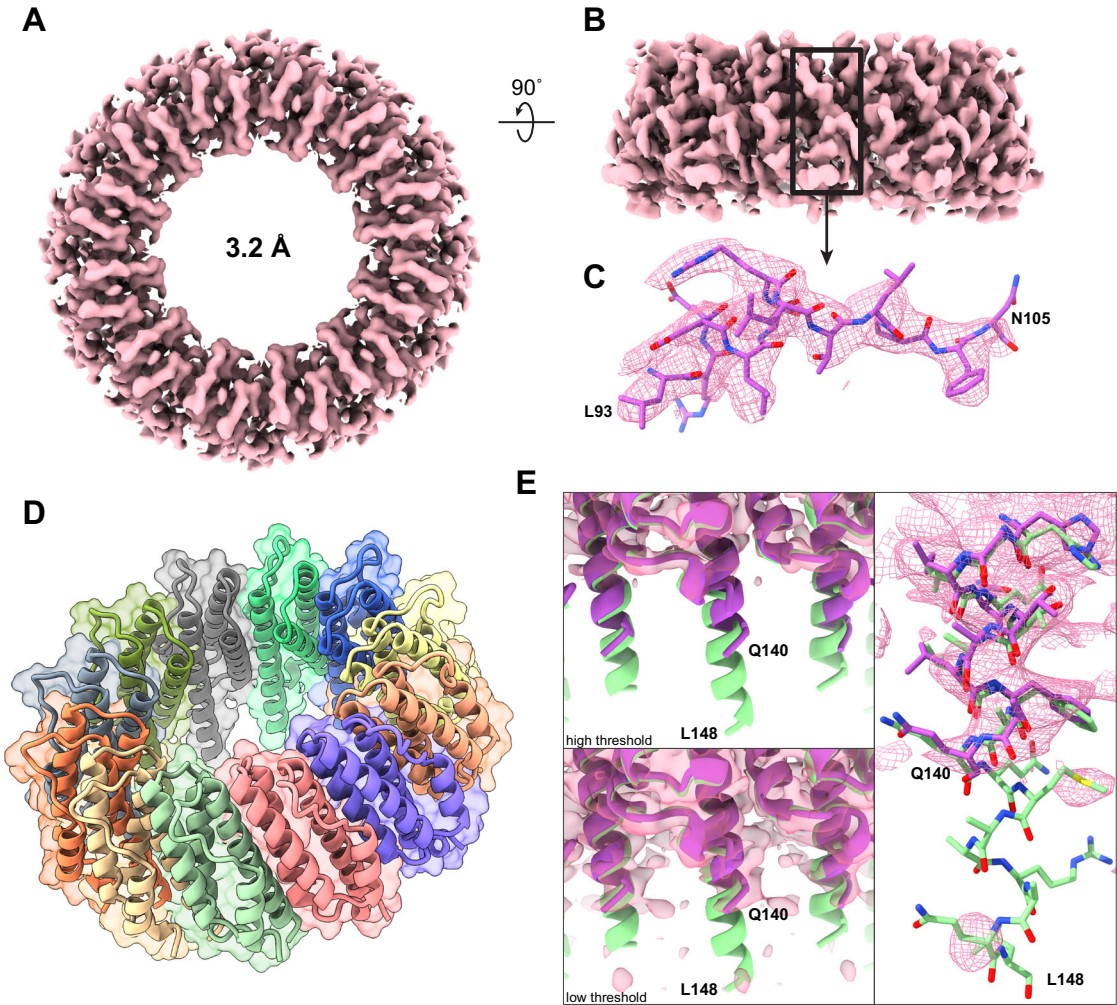

**Fig. 3 | Cryo-EM structure of full-length Bpa from *M. tuberculosis*. A** Top view of the cryo-EM map of full-length Bpa. **B** Side view of the full-length Bpa map. **C** Exemplary stretch of residues belonging to helix H3 and the H3-H4 linker of Bpa displaying confident modeling of residues into the density map. **D** Structure of the Bpa dodecameric ring. Each Bpa protomer is colored differently. **E** The crystal structure of Bpa (PDB 5LFJ, green) shows ordered C-terminal extensions of helix H4 to residue L148. The cryo-EM map of full-length Bpa (purple) is resolved up to residue Q140 indicating considerable flexibility beyond that point.

of the sample by negative stain EM (Fig. 4). In absence of substrate, Bpa$_{36-139}$ exposed to GA exhibited an elution profile identical to that of dodecameric, non-crosslinked Bpa$_{36-139}$, demonstrating that crosslinking did not affect the assembly state of Bpa$_{36-139}$ (Fig. 4A, black and red profile, respectively). In the presence of HspR, however, the primary Bpa$_{36-139}$ peak eluted slightly earlier than Bpa$_{36-139}$ alone, indicating complex formation. Furthermore, two smaller high-molecular-mass peaks were also observed. All three distinct species could be well separated and were collected as fractions I, II, and III, with fraction III being the major species (Fig. 4A). Mass spectrometric analysis of fractions II and III confirmed the crosslinked Bpa-HspR complex, as peptides corresponding to both protein components were detected (Fig. 4B).

The three fractions were analyzed via negative stain EM to assess the quality of the crosslinked particles (Fig. 4C). Fraction I featured Bpa$_{36-139}$ "filaments", most likely artifacts resulting from the protein truncation and crosslinking. Fraction II comprised Bpa$_{36-139}$ particles engaged in ring-to-ring interactions between two dodecamers (24-mers). The species found in the predominant fraction III, however, contained single, dodecameric Bpa$_{36-139}$ rings corresponding to the functional assembly state of Bpa.

Taken together, the analysis of the main fraction III showed that the Bpa$_{36-139}$-HspR$_{\Delta C9}$ complex can be stabilized upon crosslinking

with GA, and that gel filtration allows for isolation of monodisperse substrate-bound Bpa dodecamers.

To gain structural insight into the substrate recognition mechanism of Bpa, we performed single-particle cryo-EM analysis on the crosslinked Bpa$_{36-139}$-HspR$_{\Delta C9}$ complex from fraction III (Fig. 4D, Supplementary Fig. 2). Data processing with imposed C12 symmetry resulted in a 3.5 Å resolution structure of Bpa$_{36-139}$ (Fig. 4D, left panel; Supplementary Fig. 1, Supplementary Table 1). Comparing this structure to the structure of full-length Bpa determined in the absence of substrate, we could not identify significant differences within the dodecameric Bpa ring itself. To reveal additional density that may not follow the 12-fold symmetry of Bpa, we solved the non-symmetrized 3D reconstruction of the substrate-captured Bpa to a global resolution of 4.1 Å (Fig. 5A, Supplementary Fig. 1, 2, and 3, Supplementary Table 1). The non-symmetrized map clearly revealed additional density, likely arising from crosslinked HspR, within the inner cavity of the Bpa dodecameric ring. Although the pseudosymmetry and/or conformational heterogeneity of the complex prevented us from obtaining high-resolution information in this region, the extra density was localized in the proximity of the Bpa H4 helix and appeared to make contact with H131 of Bpa (Fig. 5B).

To confirm that the additional density detected within the Bpa ring originated from HspR, we employed crosslinking mass

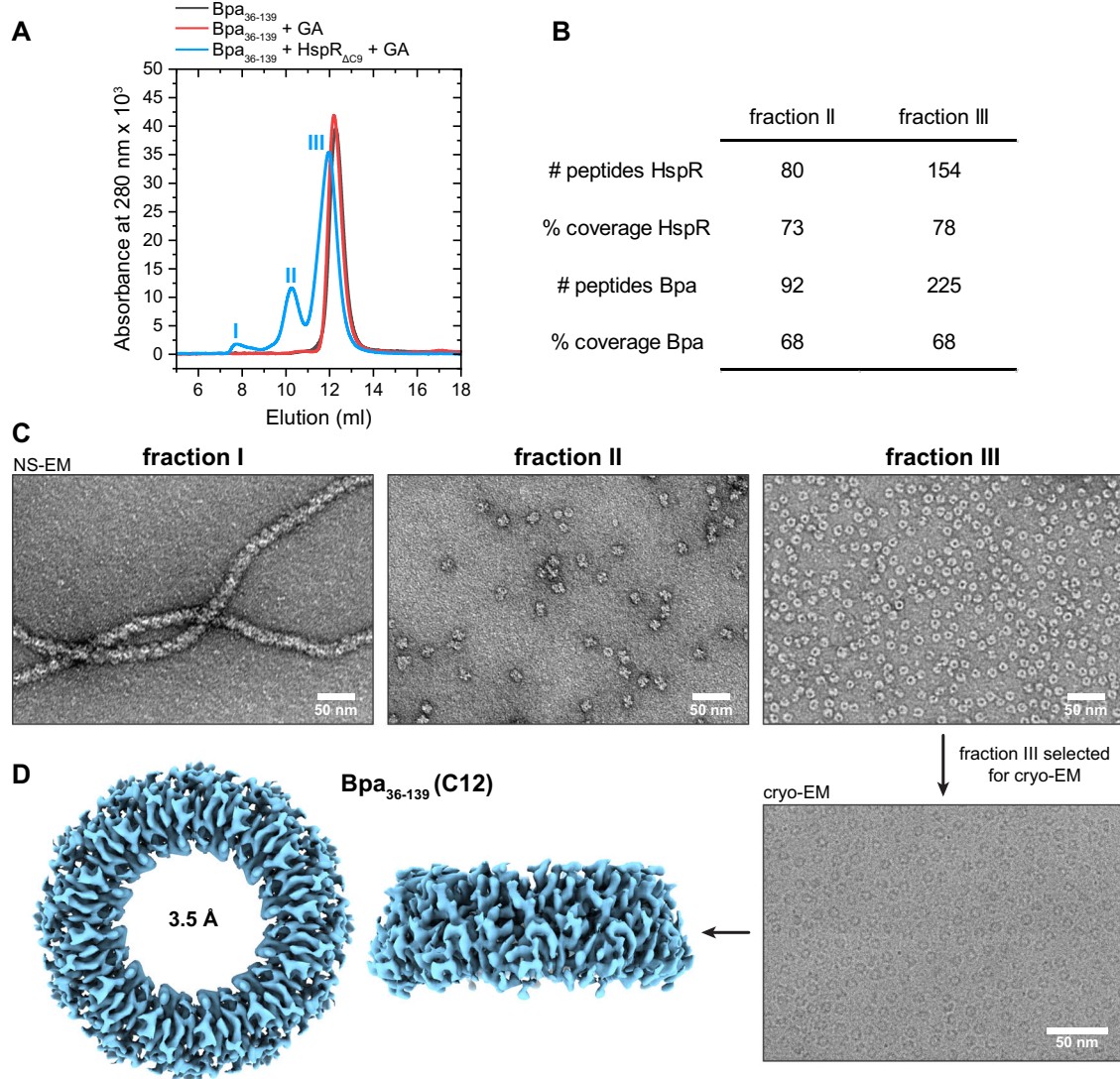

**Fig. 4 | Crosslinking and EM analysis of Bpa$_{36-139}$ in complex with HspR$_{\Delta C9}$.**
**A** Gel filtration profiles of Bpa$_{36-139}$ in absence of substrate or crosslinker (black), in the presence of crosslinker (red) and in the presence of GA crosslinker and substrate (blue). Higher molecular weight species (fractions I–III) indicated various populations of substrate-captured Bpa. **B** Mass spectrometry analysis of the crosslinked Bpa$_{36-139}$·HspR$_{\Delta C9}$ sample from **A**. Peptides for both HspR$_{\Delta C9}$ and Bpa$_{36-139}$ were found in fractions II and III. **C** Representative negative stain EM (NS-EM) micrographs of Bpa$_{36-139}$·HspR$_{\Delta C9}$ fractions from **A**. Fraction I contained Bpa$_{36-139}$ "filaments". Fraction II contained Bpa$_{36-139}$ ring-ring complexes (24-mers). Fraction III contained single Bpa$_{36-139}$ rings. **D** Cryo-EM analysis of substrate-captured Bpa$_{36-139}$ (fraction III from **A**). A representative micrograph and the high-resolution structure of Bpa$_{36-139}$ refined to 3.5 Å resolution using C12 symmetry.

spectrometry (XL-MS)[36]. In this approach, a mixture of Bpa$_{36-139}$ and HspR$_{\Delta C9}$ was subjected to crosslinking with bis[sulfosuccinimidyl] suberate (BS³). We opted for BS³ because of its homobifunctional amine-to-amine crosslinking of lysine residues, analogous to GA used for the cryo-EM samples, and because it is well-established as a crosslinking agent in XL-MS. The analysis of crosslinked peptides (Supplementary Table 2) showed that K55 of Bpa, which is the only lysine found inside the Bpa ring, formed crosslinks with HspR residues K5 and K114. These lysine residues are presumably located within the flexible N- and C-terminal regions of *M. tuberculosis* HspR, as predicted by an AlphaFold[37,38] model of the HspR dimer (Fig. 5C). The identified Bpa-HspR crosslinks are in agreement with our observation of substrate-derived densities within the cryo-EM map of the substrate-captured Bpa$_{36-139}$. The extra density protruding from Bpa residue K55 provides evidence for complex stabilization through crosslinking and verifies HspR as the ligand bound to the interior of the Bpa ring (Fig. 5B). Moreover, no crosslinks with K46 of Bpa located on the outer

surface of the Bpa ring were detected, confirming that the substrate was captured within the ring cavity as opposed to being attached in a non-specific manner to the exterior of the Bpa dodecamer.

To improve the resolution of the HspR density within the Bpa ring, we performed 3D classifications focused on the interaction site (Fig. 5D). Our analysis revealed largely different 3D class averages wherein substrate densities, despite emanating from H131 of Bpa, displayed strong heterogeneity of the surrounding substrate density. This observation would agree with a transient, highly dynamic interaction between substrate and the inner cavity of the Bpa ring.

### Bpa residues H131, F138, and R145 contribute to substrate binding

Our cryo-EM analysis revealed substrate densities near H131 of Bpa. Another study had previously proposed that Bpa residue F138 contributes to substrate binding through hydrophobic interactions[23]. Interestingly, these two residues are close to each other, separated by

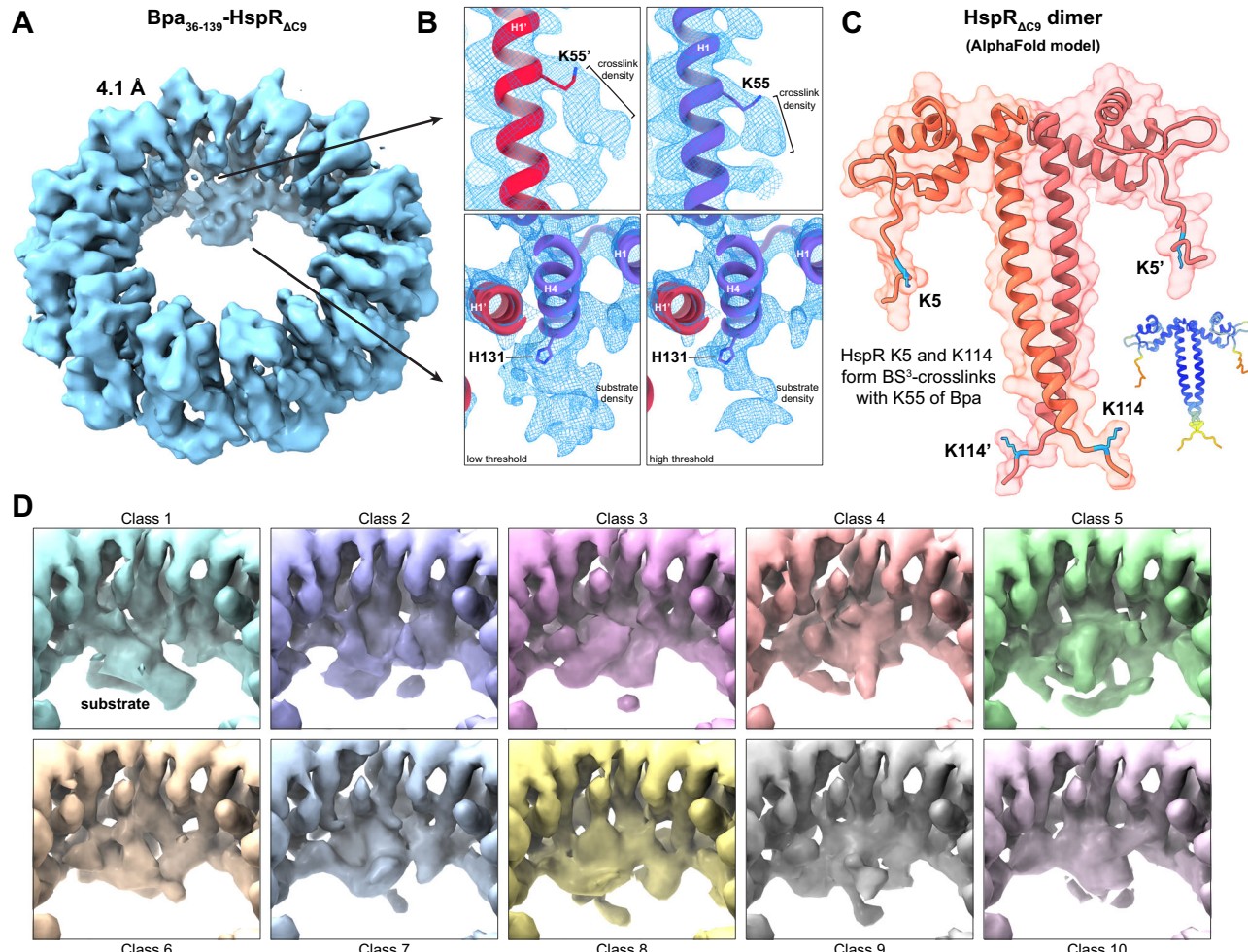

**Fig. 5 | Cryo-EM and XL-MS analyses of substrate-captured Bpa$_{36-139}$. A** Cryo-EM map of substrate-bound Bpa$_{36-139}$ solved to a global resolution of 4.1 Å. Additional density attributed to HspR$_{\Delta C9}$ is visible in the lower region of the Bpa ring. **B** Upper two panels: emerging density from Bpa residue K55 in the representative neighboring protomers (H1' in magenta, left and H1 in purple, right) corresponding to the GA crosslink formed between Bpa and HspR. Lower two panels: substrate density localized around residue H131 of Bpa H4 helix from one protomer (purple) shown at low map threshold (left) and high map threshold (right). **C** XL–MS analysis of Bpa$_{36-139}$ in complex with HspR$_{\Delta C9}$. Lysine residues of HspR$_{\Delta C9}$ cross-linked to Bpa$_{36-139}$ are mapped onto the AlphaFold model of the HspR$_{\Delta C9}$ dimer. A downsized copy of the model colored according to the per-residue confidence score is shown on the right (dark blue – high confidence; red – low confidence). **D** Focused 3D classification revealing considerable heterogeneity within the Bpa-HspR interaction site and demonstrating multiple conformational states of the substrate around H131 of Bpa.

only two helical turns with their aromatic side chains pointing into the Bpa cavity in the published crystal structures (PDB 5LFJ, 5LFQ, 5IET)[21,22]. Although F138 could not be resolved in our cryo-EM map of the Bpa$_{36–139}$-HspR$_{\Delta C9}$ complex, the HspR density found in multiple 3D class averages (Fig. 5D) indeed extended downward and encompassed regions that would include F138. We hypothesized that H131 and F138 interact with residues of the HspR DNA-binding domain through ring-stacking contacts. Additionally, the last resolved residue within the crystal structure, R145, is situated in close proximity below F138 and could potentially form electrostatic interactions with HspR.

To investigate the importance of these Bpa residues in substrate recruitment, we substituted H131, F138, and R145, one at a time, with serine residues. Our rationale was that the short, uncharged serine side chain would not engage in ring stacking or electrostatic interactions with HspR. To ensure that any impairment in HspR degradation was not due to a loss of structure in Bpa, we recorded CD spectra for all generated Bpa variants and found that, like wild-type Bpa, they exhibited a CD signature with the characteristic α-helical secondary structure contribution (Supplementary Fig. 5).

Using in vitro degradation assays, we demonstrated that all three single mutants of Bpa slowed down proteasomal degradation of HspR compared to wild-type Bpa (Fig. 6A, B). Our gel densitometry analysis showed that ~80% of HspR molecules were degraded by the proteasome within 15 minutes in the presence of wild-type Bpa, whereas only around ~50–55% of the substrate was degraded for H131S, F138S, or R145S Bpa variants within the same time frame. In addition to individually mutating the putative substrate binding residues, we also assessed the combined effect of H131S, F138S, and R145S substitutions on HspR recruitment. The corresponding double (H131S-F138S) and triple mutant (H131S-F138S-R145S) of Bpa exhibited severe impairment in HspR degradation, with only around 25 and 20% of HspR molecules degraded after 15 minutes, respectively (Fig. 6A, B). This is similar to the control experiment in the absence of Bpa, which showed that the 20S CP alone could degrade around 12% of HspR within the same time frame.

As we hypothesized that Bpa residues H131, F138, and R145 are involved in substrate binding (and not e.g., Bpa anchoring to the proteasome), we investigated the binding of Bpa variants to HspR (Fig. 6C). We fluorescently labeled wild-type Bpa, H131S, R145S, H131S-F138S double mutant, and H131S-F138S-R145S triple mutant and monitored the temperature-related fluorescence intensity changes of these proteins at increasing concentrations of added HspR. Wild-type

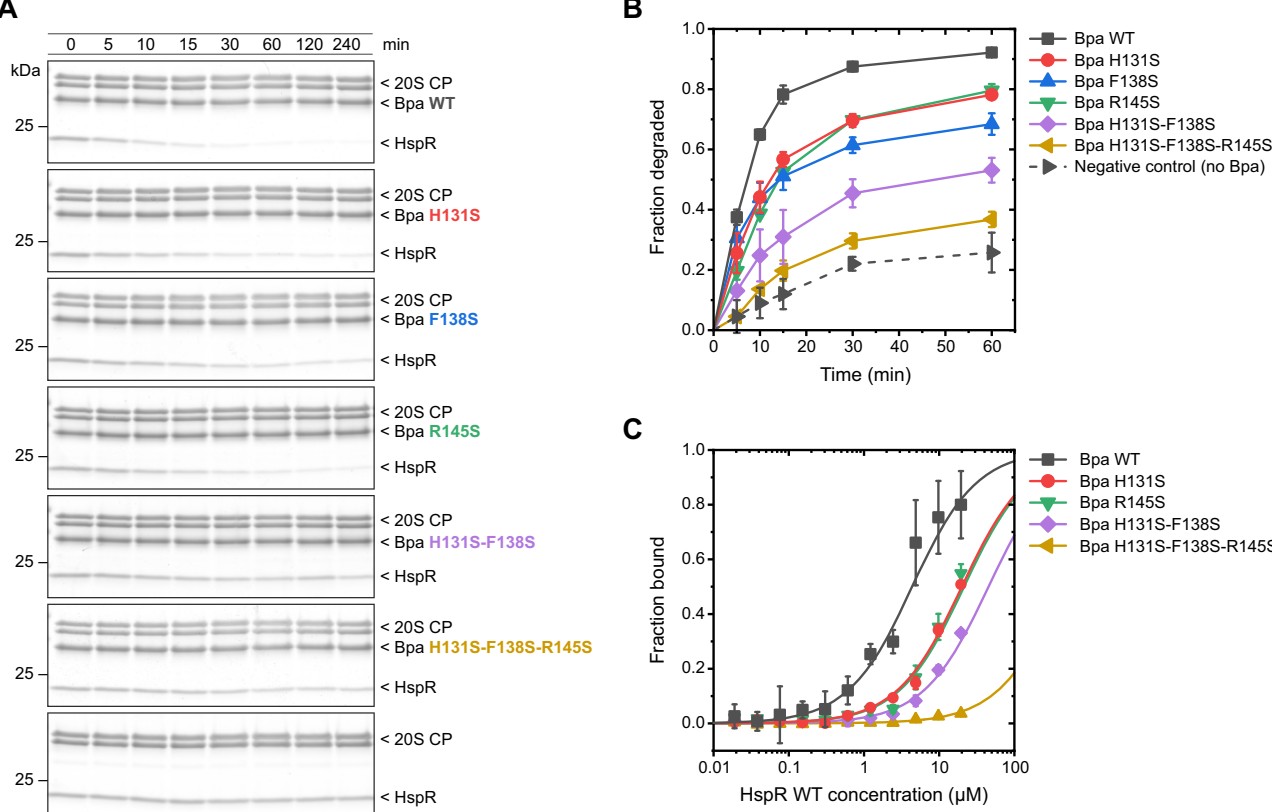

**Fig. 6 | Bpa residues H131, F138, and R145 are involved in HspR recruitment to Bpa-mediated proteasomal degradation. A** In vitro degradation assays of wild-type HspR using site-directed variants of Bpa, followed by Coomassie-stained SDS-PAGE. **B** Progress of degradation was measured by following the change in the HspR gel band intensity shown in **A**. Densitometric analysis was conducted from gel triplicates and data are presented as mean ± standard deviation in gel band intensity. **C** Binding of wild-type HspR to fluorescently labeled Bpa variants was measured by recording the temperature-related fluorescence intensity changes at different concentrations of substrate. Reactions were run in triplicates and dependency of bound Bpa fraction on the substrate concentration was fitted with the solution to the quadratic equation for the fraction of fluorescent molecules that formed the complex. Data are presented as mean ± standard deviation.

Bpa bound HspR with a dissociation constant ($K_D$) of 4.3 ± 0.98 μM, which is comparable to the $K_D$ reported previously[28]. The other Bpa variants bound HspR with much lower affinities, which could not be determined reliably due to the low solubility of HspR (Fig. 6C). Together, our results demonstrate an important, and likely cooperative role of the Bpa residues H131, F138, and R145 in the initial binding of Bpa to its natural substrate HspR.

## Discussion

ATP-independent proteasomal degradation is a pathway that operates in both eukaryotes and bacteria alongside the energy-dependent proteasomal degradation routes. In eukaryotes, the known ATP-independent proteasomal activators (PA200 and PA28) recruit highly disordered substrates to the 20S CP in a ubiquitin-independent manner[26,39–43]. For example, PA200 was shown to stimulate the proteasomal degradation of the non-native tau protein and of acetylated histones[26,42–44]. Likewise, PA28γ was shown to facilitate the proteasomal degradation of non-native substrates[25,45–47], including human tumor suppressor p52 and cyclin kinase inhibitor p21[40]. Although there have been numerous studies focusing on the association of ATP-independent activators with the 20S CP in eukaryotes, the substrate recruitment mechanisms in these systems remain poorly understood.

In bacteria, ATP-independent proteasomal degradation is facilitated by Bpa, and only a limited number of Bpa-dependent proteasomal substrates have been identified[19,20]. Our previous work introduced a model for the recruitment mechanism underlying Bpa-mediated proteasomal degradation of HspR in mycobacteria. According to this model, the N-terminal HTH domain of HspR interacts with Bpa, which

subsequently enables the insertion of substrate into the proteasomal degradation chamber via its C-terminal unstructured tail[28]. In the present study, we build on this understanding of substrate recruitment by examining the process from the perspective of Bpa. To achieve this, we determined the structure of Bpa in complex with HspR and conducted in vitro biochemical experiments to provide further insight into the mechanism.

Our structural analysis revealed substrate densities near residue H131 inside the Bpa ring cavity. We therefore postulated that H131, along with F138 previously suggested to be involved in substrate binding[23], and the nearby R145 are likely involved in the initial recruitment of HspR to Bpa. Considering our previous study that implicated HspR residue H24 in the substrate binding to Bpa[28], it is conceivable that Bpa interacts with HspR through ring stacking of aromatic residues involving H24 on HspR and H131 and potentially F138 of Bpa.

Moreover, electrostatic interactions are likely to play a role in the substrate selection process. We previously showed that HspR residue E19 is involved in the binding to Bpa[28], the counterpart of which could be R145 of Bpa identified in this study. This hypothesis is supported by the fact that R145 is located in close proximity to residue F138 with its side chain pointing into the Bpa cavity according to the crystal structure. To validate our interaction model, we mutated H131, F138, and R145 within full-length Bpa and observed that each of the single mutants slowed the degradation of HspR compared to wild-type Bpa. Notably, the triple mutant exhibited the most pronounced effect on HspR degradation, implying an additive effect of these residues (Fig. 6). In conclusion, our findings support a

model wherein three closely positioned Bpa residues H131, F138, and R145 situated vertically along the helix H4 extensions, mediate substrate engagement through a combination of ring-stacking and charge-charge interactions.

We have previously shown that HspR engages with Bpa through its predicted N-terminal HTH domain[28]. In this study, we observed contact between Bpa residue H131 (Fig. 5B) and the substrate density. The heterogeneity surrounding the substrate density adjacent to Bpa residue H131 suggests that HspR might undergo an order-to-disorder transition upon binding Bpa or that HspR segments beyond this point of engagement are flexible. While we showed previously that HspR in isolation maintains a folded conformation[28], it is conceivable that a local concentration of surface residues within the Bpa ring could destabilize HspR upon binding to Bpa. Furthermore, the measured width of the AlphaFold-predicted HspR dimer is more than 60 Å, whereas the Bpa pore is only 40 Å wide at the top and narrows further towards the helix H4 extensions. Consequently, the entry of a fully folded HspR dimer into the Bpa funnel seems unlikely. We propose that the dynamic monomerization-dimerization and/or folding-unfolding of an HspR dimer allows HspR to enter Bpa in a monomeric state, potentially already partially disordered, while maintaining a certain level of structure required for the initial Bpa binding. This aligns with our previous findings that disorder is required but not solely sufficient for Bpa-mediated proteasomal degradation[28]. In this context, the asymmetric tethering of seven adjacent Bpa protomers into the pockets of the proteasome α-ring subunits, along with the observed hinged upward movement of the Bpa ring, presents intriguing functional possibilities. Movements of Bpa on the proteasome α-ring could play a role in the process of substrate translocation by either a rolling-wobbling movement relative to the platform of the proteasome or simply a hinging up and down motion. However, further studies into the dynamics of the complex would be required to address these questions.

Our study offers structural and mechanistic insights into the ATP-independent substrate recruitment by the Bpa activator for proteasomal degradation in mycobacteria. Using cryo-EM, we solved structures of Bpa with and without the proteasome allowing us to observe considerable flexibility within the C-terminal stretch of Bpa, which forms additional contacts with the proteasomal α-subunits beyond the GQYL motif. Furthermore, through a combination of structural analyses and in vitro experiments, we identified specific Bpa residues responsible for recruitment of HspR into the Bpa ring cavity. Our results provide a structural framework for Bpa-dependent substrate recruitment to proteasomal degradation and provide intriguing hypotheses regarding substrate processing to be explored in future studies.

## Methods

### Cloning and protein purification
Genes for all proteins were amplified from *M. tuberculosis* H37Rv DNA by PCR. Coding sequences for Bpa and HspR variants were cloned into the pET28a vector under the T7 promoter by Gibson assembly. Coding sequences for proteasome variants were cloned into a pETDuet vector also under the T7 promoter using Gibson assembly. For proteasome constructs, the β-subunit sequence lacked the propeptide sequence and a C-terminal Strep-tag was added. Constructs were transformed into *E. coli* Rosetta cells and over-expressed in autoinduction medium[48].

Proteasome constructs were purified by affinity chromatography using Strep-TactinXT resin (IBA Lifesciences) and proteasome-containing fractions were applied to a Superose 6 gel filtration column equilibrated in 50 mM HEPES-KOH, pH 7.5, 150 mM NaCl, 10% (v/v) glycerol, and 1 mM EDTA. His$_6$-MBP-tagged HspR and His$_6$-Bpa constructs were isolated by Ni-NTA affinity chromatography. Bpa-containing fractions were loaded on a Superdex 200 pg (Cytiva) gel filtration column equilibrated in 50 mM HEPES-KOH, pH 7.5, 150 mM NaCl, 10% (v/v) glycerol, and 1 mM EDTA. His$_6$-MBP-tagged HspR-containing fractions were loaded on a Superdex 75 pg (Cytiva) gel filtration column equilibrated in 50 mM HEPES-KOH, pH 7.5, 150 mM NaCl, 5% (v/v) glycerol, and 5 mM MgCl$_2$.

HspR (wild-type HspR and HspR$_{\Delta C9}$) was purified from inclusion bodies under denaturing conditions as previously described[28]. In brief, cell lysates were solubilized in 60 mM EDTA, 1.5 M NaCl, 6% (v/v) Triton X-100 and inclusion bodies were harvested by centrifugation (47 850 x g). Inclusion bodies were washed in 100 mM HEPES-KOH pH 7.5, 20 mM EDTA before solubilization in 50 mM HEPES-KOH pH 7.5, 6 M guanidinium chloride. Cell debris was removed by ultracentrifugation (100,000 × g) before refolding of the solubilized inclusion bodies by dialysis against 50 mM HEPES-KOH, 0.5 M arginine pH 7.5. Refolded protein was further dialyzed against 50 mM HEPES-KOH pH 7.5, 50 mM NaCl, 5 mM MgCl$_2$, 5% (v/v) glycerol and purified over a heparin column using a 50 mM to 2 M NaCl linear gradient (in 50 mM HEPES-KOH pH 7.5, 5 mM MgCl$_2$, 5% (v/v) glycerol). Lastly, HspR was further purified by gel filtration with a Superdex 75 column (Cytiva) equilibrated in 50 mM HEPES-KOH pH 7.5, 150 mM NaCl, 5 mM MgCl$_2$, 5% (v/v) glycerol.

### Gradient fixation and batch crosslinking
To study the Bpa-CP complex by EM, the sample was purified over a linear 10-30% (w/v) sucrose gradient in 50 mM HEPES-KOH pH 7.5, 150 mM NaCl, 5 mM MgCl$_2$, and 5% (v/v) glycerol, with the addition of 0.15% (v/v) GA to the 30% (w/v) sucrose buffer[34,35]. 1.6 μM assembled full-length 20S proteasome (β-subunit T1A inactive variant) was incubated for 30 min at room temperature with 4 μM Bpa$_{12}$ in 50 mM HEPES-KOH pH 7.5, 150 mM NaCl, 5 mM MgCl$_2$, 5% (v/v) glycerol before applying the sample on the gradient. Gradients were then ultra-centrifuged at 109 400 x g for 16 hours at 4 °C (Beckman Coulter). 400 μL gradient fractions were collected and visualized with SDS-PAGE after the crosslinking was quenched by addition of 50 mM Tris-HCl, pH 8.0. 100 μM of MBP-tagged HspR was only added to the assembled Bpa-CP complex after buffer was exchanged by applying the assembled complex onto a PD10 column (Cytiva) and eluting with 50 mM HEPES-KOH pH 7.5, 150 mM NaCl.

To investigate the interaction between Bpa and its substrate HspR, we performed batch crosslinking between Bpa$_{36-139}$ and HspR$_{\Delta C9}$. 5 μM Bpa$_{12}$ was incubated with 30 μM HspR$_{\Delta C9}$ dimer in the presence of 0.1% (v/v) GA in 50 mM HEPES-KOH pH 7.5, 150 mM NaCl for 30 min at room temperature. The reaction was quenched by addition of 50 mM Tris-HCl pH 8.0. Protein complexes were then loaded onto a Superdex 200 10/300 (Cytiva) analytical gel filtration column equilibrated in 50 mM HEPES-KOH pH 7.5, 150 mM NaCl to isolate different crosslinked species.

### Negative stain EM sample preparation and data collection
To visualize different species of Bpa$_{36-139}$ batch-crosslinked with GA to HspR$_{\Delta C9}$ as well as Bpa-CP complexes prepared via gradient fixation with GA, we performed negative stain EM. Each sample, i.e. Bpa$_{36-139}$–HspR$_{\Delta C9}$ species corresponding to the first, second, or third gel filtration peak (elution volume of 7.8 mL, 10.3 mL, and 12 mL, respectively) and different Bpa-CP complexes, was prepared following the same procedure. Specifically, 4 μL sample was applied to glow-discharged (PELCO easiGlow, Ted Pella, negative, 25 mA, 30 s) carbon-supported Quantifoil grids (400 mesh) and incubated for 1 min, followed by blotting with a filter paper, washing in two droplets of water, and sequential staining in two droplets of 2% (w/v) aqueous uranyl acetate. After 1–2 min incubation, the grid was blotted and air-dried[49]. The grids were imaged with Morgagni 268 (100 kV) transmission electron microscope (Thermo Fisher Scientific). For automated data collection, Talos L120C G2 (120 kV) transmission electron microscope, equipped with the Ceta S 16 M 4k×4k CMOS detector (Thermo Fisher Scientific), was used. Images were recorded with the EPU software

(Thermo Fisher Scientific) as single micrographs at a dose of approximately 30 e⁻/Å² with a resulting pixel size of 1.9 Å/pixel at 73'000-fold magnification. The targeted defocus was set in the range of −2.5 to −1.0 μm with 0.3 μm increments.

## Negative stain EM single-particle analysis

Negative stain EM datasets were processed in cryoSPARC[50]. For each dataset, the Patch CTF Estimation routine was used to estimate the CTF. Then, 100-300 particles were manually selected, extracted, and 2D classified. The resulting 2D class averages were then used as references for automated particle picking from all the micrographs using the cryoSPARC Template Picker. The identified particles were extracted with a box size of 240 pixels and binned to 4.56 Å/pixel, before being subjected to multiple rounds of 2D classification to remove false-positive picks. Particles corresponding to the good 2D class averages were then used to generate ab initio models. Finally, the 3D map exhibiting strong density for both CP and Bpa served as a reference in 3D refinement (cryoSPARC homogeneous refinement or non-uniform refinement).

## Cryo-EM sample preparation and data collection

To attain the cryo-EM structure of full-length Bpa, 3.6 μL of 5 μM Bpa₁₂ in 50 μM HEPES-KOH, pH 7.5, 150 mM NaCl, and 0.003% (v/v) NP40 were added to R2/2 holey carbon copper grids (Quantifoil) and plunge-frozen. To solve the structure of a substrate-captured Bpa, the Bpa₃₆₋₁₃₉ − HspR$_{\Delta C9}$ complex was stabilized via crosslinking and isolated by gel filtration as described above. Then, 3.6 μL of sample were added to R2/2 holey carbon copper grids, coated with a thin carbon layer (prepared in-house, 1–1.5 nm) and let to adsorb for 1 min before plunge-freezing. For Bpa in complex with the proteasome, 3.6 μL of the 20S Bpa-CP/HspR sample were added to Quantifoil R2/2 holey carbon copper grids and plunge-frozen. Plunge-freezing for all samples was done in a liquid ethane/propane mix using a Vitrobot Mark IV (Thermo Fisher Scientific).

The cryo-EM data were collected on Titan Krios TEM microscopes at 300 kV (Thermo Fisher Scientific), equipped with K3 direct electron detectors operating in counting mode (Gatan) using a slit width of 20 eV on a GIF-Quantum energy filter. Datasets of 22'538, 27'509, and 30'384 movies were collected for 20S Bpa-CP/HspR (0.66 Å/pixel), full-length unbound Bpa (0.51 Å/pixel), and Bpa₃₆₋₁₃₉ complexed with HspR$_{\Delta C9}$ (0.51 Å/pixel), respectively, using EPU software (Thermo Fisher Scientific). For all datasets, a similar dose of approximately 80 e⁻/Å², and the −2.6 to 1.2 μm range of targeted defocus with 0.2 μm increments, were used.

## Cryo-EM single-particle analysis

Initial processing steps were identical for all cryo-EM datasets recorded in this study. First, motion-corrected and dose-weighted[51] movies were imported into cryoSPARC[50] where CTF, astigmatism, and relative ice thickness were estimated using the Patch CTF Estimation routine. Subsequent to removal of poor-quality micrographs, approximately 200 particles were manually picked and 2D classified to generate templates which were later used for the automated particle-picking from 500 randomly selected micrographs (Template Picker). Following particle extraction and 2D classification, 3000 particles were selected to train the Topaz model[52,53]. Particles were then picked from all micrographs using the trained model, extracted, and 2D classified to remove false-positive picks.

For the 20S Bpa-CP/HspR dataset, ~305,000 particles (320-pixel box size; 1.32 Å/pixel) were processed. First, an ab initio model created in cryoSPARC (C1, 3 classes) served as a reference in the heterogeneous 3D refinement with 4 classes and default parameters. After selecting 3D class averages of good quality with a strong signal for CP and both Bpa rings (~129,000 particles), a homogeneous 3D refinement, with imposed C2 symmetry and a mask covering CP and both Bpa caps, was

performed. The resulting map at 3.3 Å global resolution was then filtered according to their local resolution estimates (cryoSPARC).

The map of the 20S CP without Bpa was generated via signal subtraction of the dataset described above. All particles aligned during the heterogeneous refinement (~305,000 particles) were imported[54] into Relion[55] where they were re-extracted (400-pixel box size, 1.06 Å/pixel) and locally 3D refined (mask over full complex) to the respective global resolution of 3.3 Å. The signal subtraction was performed on the refined particles by applying a mask on CP and shifting the new box (300 pixels) to the center of that mask. The new particle stacks were then imported back into cryoSPARC where they were subjected to heterogeneous 3D refinement with 4 classes and imposed D7 symmetry. A single 3D class average of the highest estimated resolution (~154,000 particles) was selected and used in final non-uniform 3D refinement (D7) with defocus and CTF optimization to give a map at 2.5 Å global resolution. The final map was filtered according to the local resolution estimates (cryoSPARC).

The full-length unbound Bpa dataset was processed entirely in cryoSPARC with imposed C12 symmetry. The 300'000 out of 2.1 million particles (100-pixel box size; 2.04 Å/pixel) selected after 2D classification were used to generate three ab initio models. The highest quality map was then used as a reference in a heterogeneous 3D refinement with 6 classes. Approximately 850'000 particles, corresponding to the single 3D class average of the highest estimated resolution, were selected and subjected to a non-uniform 3D refinement with dynamic masking which gave a map at 4.3 Å resolution, limited by the pixel size. After re-extracting the particles (200-pixel box size; 1.02 Å/pixel), another heterogeneous 3D refinement with 4 classes was performed. Finally, a single class of ~230'000 particles reconstructed to the estimated 3.6 Å resolution was selected for the final non-uniform 3D refinement using dynamic masking to give a map at 3.2 Å global resolution. The map was filtered based on local resolution estimates (cryoSPARC).

The dataset of Bpa₃₆₋₁₃₉ crosslinked with HspR$_{\Delta C9}$ was processed in cryoSPARC, Relion, and cisTEM[56]. Following 2D classification, ~4.3 million particles (100-pixel box size; 2.04 Å/pixel) were subjected to heterogeneous 3D refinement with 6 classes using the low-pass filtered map (15 Å) of full-length Bpa-apo 3D reconstruction as a reference. Approximately 3.7 million particles corresponding to the good 3D class averages (better than estimated 5 Å resolution) were selected and refined using non-uniform 3D refinement yielding a 4.3 Å resolution 3D reconstruction (limited by the pixel size). The refined particles were then re-extracted with a box size of 200 pixels (1.02 Å/pixel) before another non-uniform 3D refinement, using the previous map as a reference and a mask covering the Bpa ring along with the inner cavity (created in UCSF Chimera[57] and cryoSPARC), generated a map at 3.8 Å resolution. Subsequently, the map and refined particles were once again used in heterogeneous 3D refinement with 10 classes. Particles corresponding to the classes where the well-resolved Bpa ring contained additional density within its inner cavity were combined (~1.1 million particles) and refined to 3.8 Å resolution using non-uniform 3D refinement with masking over Bpa ring and inner cavity. Although the Bpa 12-mer could be resolved to high resolution, particle heterogeneity and pseudosymmetry made further attempts at resolving the substrate density within the inner cavity of Bpa using cryoSPARC, unsuccessful. Therefore, particles were imported into Relion and 3D refined with global searches and a mask including the ring and the cavity to a resolution of 4.1 Å. The refined particle stack was then imported into cisTEM and subjected to auto-refinement with global searches and a spherical mask of 130 Å in diameter. The resulting map at 4.2 Å resolution revealed substrate densities within the inner cavity of the Bpa ring. Further two rounds of 3D classification (without alignments) focused on the substrate density allowed the separation of different classes, where the substrate seemed to bind to neighboring Bpa subunits. Next, the selected ~470'000 particles were imported

back into cryoSPARC where local refinement was performed using 5° and 5 Å searches. The resulting map at 4.1 Å global resolution was filtered based on local resolution estimates (cryoSPARC). Unfortunately, subsequent no-alignment 3D classifications with a mask applied only on the substrate density demonstrated a large degree of heterogeneity hindering high-resolution structure determination.

## Model building and refinement

The structure of the *M. tuberculosis* 20S CP (PDB ID: 5LZP) was rigid-body fitted to our cryo-EM map solved for the full-length 20S CP using UCSF Chimera. The asymmetric unit of the proteasome comprising one α- and one β-subunit along with a single GQYL motif of Bpa was extracted (The PyMOL Molecular Graphics System, Version 2.4 Schrödinger, LLC) and subjected to five cycles of real space refinement in Phenix[58,59] with standard parameters, using protein secondary structure and side chain rotamer restraints. After manual real space refinement in Coot[60] was used to reduce clashscore and rotamer outliers, as well as N-terminally elongating the GQYL motif by two residues, the model was once again refined in Phenix using the same parameters as described above. The 20S CP was then reconstructed by applying D7 symmetry before final real space refinement imposing non-crystallographic symmetry (NCS), protein secondary structure and side chain rotamer restraints.

For the full-length unbound Bpa model, the structure of the *M. tuberculosis* Bpa dodecamer (PDB ID: 5LFJ) was rigid-body fitted to our cryo-EM map using UCSF Chimera. A single Bpa subunit was then extracted and C-terminally truncated to Q140 using PyMOL, as no density beyond that residue was resolved in our map. Next, the Bpa chain underwent five cycles of real space refinement in Phenix with default parameters, where protein secondary structure and side chain rotamer restraints were used. Manual real-space refinement in Coot was done to reduce the clashscore and rotamer outliers before the model was subjected to another real-space refinement in Phenix using the same parameters. The Bpa dodecamer was then reconstructed by applying C12 symmetry to the refined model, followed by the final real space refinement imposing NCS, secondary structure, and side chain rotamer restraints.

For the Bpa$_{36-139}$ model, the structure solved for Bpa apo was rigid-body fitted to our cryo-EM map using UCSF Chimera. Due to weaker densities observed around the termini of the model, a single Bpa chain was truncated to the range of residues 39–133 and subjected to five cycles of real space refinement in Phenix using protein secondary structure and side chain rotamer restraints. Following manual real-space refinement in Coot to minimize clashscore and rotamer outliers, the model was again real-space refined with identical parameters. Next, the truncated Bpa dodecamer was generated by imposing C12 symmetry and underwent final real-space refinement in Phenix imposing NCS, secondary structure, and side chain rotamer restraints. The resulting model was also refined against the cryo-EM map reconstructed in C1 symmetry using the same parameters as above but excluding NCS restraints.

The comprehensive validation tool in Phenix was used to validate refined models and MolProbity[61] generated model statistics. All cryo-EM maps and atomic models were visualized with UCSF ChimeraX[57].

## Proteasomal degradation assays

The Bpa-dependent proteasomal degradation of HspR was carried out at 37 °C in 50 mM HEPES-KOH pH 7.5, 150 mM NaCl, 5 mM MgCl$_2$, 1 mM DTT. HspR (4 µM protomer) was incubated in the presence of 1 µM Bpa$_{12}$ and 0.4 µM assembled proteasome. Samples were taken at the indicated time points and quenched in Laemmli loading buffer. Substrate degradation in the form of decreased HspR gel band intensity was followed via SDS-PAGE and densitometry (GelAnalyzer 23.1.1). To account for variability between lanes, the HspR gel band intensity was normalized with respect to the proteasomal α- and β-subunit bands

within the same lane. Each experiment was carried out in triplicate and the presented graph shows their respective mean and standard deviation.

## Substrate binding assays

To measure the affinity of Bpa variants to HspR, temperature-related fluorescence intensity changes were recorded on a Monolith NT.115 from NanoTemper. First, Bpa lysine residues were labeled with three-fold excess of RED-NHS second-generation dye using the RED-NHS second-generation kit according to manufacturer guidelines (NanoTemper). 250 nM Bpa-RED protomer was titrated with serial dilutions of 20 µM to 10 nM of wild-type HspR. Reactions were pre-incubated for 10 min at room temperature and aspirated into standard capillary tubes (NanoTemper). The data were recorded at 25 °C with 20% red LED power and 40% MST power. Scans were compiled and analyzed with the NTAffinity Analysis v2.0.2 software, extracting the dissociation constant by fitting the binding curve with the quadratic equation solution for the fraction of fluorescent molecules that formed the complex[62].

## Circular dichroism

To analyze the protein fold of Bpa, we used circular dichroism (CD) to measure secondary structure content. Bpa variants were dialyzed into 20 mM Na/K phosphate buffer (pH 7.5) and CD spectra were recorded by measuring ellipticity (θ) in a 0.1 cm quartz cuvette (Hellma Analytics) at 0.2 mg/mL placed in a Jasco J-710 Spectropolarimeter (Brechbühler). The measured ellipticity (θ) from five acquisitions was normalized to mean molar ellipticity in degree cm$^2$ dmol$^{-1}$:

$$MRW = \frac{\theta * 100 * kDa}{c * d * N}$$

Where $N$ is the number of amino acid residues, $d$ is the cuvette pathlength in cm, and $c$ is the protein concentration in mg/mL.

## Crosslinking mass spectrometry

For mass spectrometry analysis of crosslinking sites, the Bpa-HspR complex (45 µM Bpa$_{36-139}$ monomer and equimolar amounts of HspR$_{ΔC9}$ monomer) was crosslinked with 1 mM BS[3] for 30 min at 25 °C, and the reaction was stopped by addition of 50 mM Tris-HCl pH 8.0. The crosslinked complex was purified by gel filtration on a Superdex 200 10/300 (Cytiva) analytical column equilibrated in 50 mM HEPES-KOH pH 7.5, 150 mM NaCl. The fraction corresponding to the cross-linked sample was collected and evaporated to dryness in a vacuum centrifuge. The dried residue was dissolved in 8 M urea and tris(2-carboxyethyl) phosphine was added to a final concentration of 2.5 mM. Cysteine thiols were reduced by incubation at 37 °C for 30 min. Alkylation of free thiol groups was achieved by the addition of iodoacetamide to 5 mM final concentration and incubation for 30 min at room temperature, protected from light. Enzymatic digestion was performed in two steps: First with endoproteinase Lys-C (Wako, 1:100 enzyme:substrate ratio) after dilution to 5.5 M urea and incubation for 3 h at 37 °C; then with trypsin (Promega, 1:50 enzyme:substrate ratio) after further dilution to 1 M and incubation at 37 °C overnight. The digest solution was acidified by addition of formic acid to 2% (v/v), and the sample was desalted by solid-phase extraction using a SepPak tC18 cartridge (Waters), followed by fractionation by gel filtration (Superdex 30 Increase, Cytiva)[36,63].

Four gel filtration fractions were analyzed in duplicate by liquid chromatography-tandem mass spectrometry on a system consisting of an Easy nLC-1200 HPLC instrument coupled to an Orbitrap Fusion Lumos mass spectrometer (both Thermo Fisher Scientific). Each analysis consisted of a 60-min gradient separation on an Acclaim PepMap RSLC C$_{18}$ column (250 mm × 75 µm, Thermo Fisher Scientific). The

gradient was set from 11 to 40% B with mobile phases A = water/acetonitrile/formic acid (98:2:0.15, v/v/v) and B = acetonitrile/water/formic acid (80:20:0.15, v/v/v). The flow rate was 300 nL/min.

Mass spectrometry was performed in data-dependent acquisition mode with a cycle time of 3 s (top speed mode). MS spectra were acquired in the orbitrap analyzer at a resolution of 120'000, and candidate precursors were picked if they had a charge state between +3 and +7 and were not on an exclusion list (dynamic exclusion window = 30 s). Fragmentation was performed in the linear ion trap with a normalized collision energy of 30%, and fragments were detected in the orbitrap at a resolution of 30'000.

Crosslinked peptides were identified from MS data using xQuest (version 2.1.5)[63,64] against a database containing the sequences of the target proteins Bpa and HspR and a human keratin (UniProt K2C1_HUMAN) identified as a contaminant. Protease specificity was set to cleavage after lysine and arginine unless followed by proline, with a maximum of two missed cleavages per peptide allowed (excluding the crosslinking site). Precursor and fragment mass errors were initially set to ±15 ppm, and the precursor mass window was further adjusted to +1 to +6 ppm post-search. All identifications were further filtered for a % TIC (total ion current) value ≥ 0.15 and a minimum number of bond cleavages of 5 per peptide chain. A target/decoy search using reversed sequences of the database entries was used to estimate the false discovery rate, and no decoy hits were identified with the selected score threshold of 25.

### Reporting summary
Further information on research design is available in the Nature Portfolio Reporting Summary linked to this article.

## Data availability
The structural data reported in this study are available in the Electron Microscopy Data Bank and Protein Data Bank under accession codes EMD-19150 (20S Bpa-CP), EMD-19151 (20S CP), EMD-19162 (full-length Bpa), EMD-19159 and 19153 (Bpa$_{36-139}$–HspR$_{\Delta C9}$), and PDB ID 8RGX (full-length Bpa). Mass spectrometry data has been uploaded to PRIDE (Project accession: PXD047368). Source data are provided with this paper.

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

## Acknowledgements

We would like to thank Paul Bachman from the Glockshuber group at ETH Zurich for IT support and help with initial screening. The cryo-EM data were collected at the Scientific Center for Optical and Electron Microscopy at the ETH Zurich (ScopeM). We thank the ScopeM staff, especially M. Peterek, for technical support during data collection. We also thank the staff of the Functional Genomics Center Zurich for confirmation of protein identities in this study. This work was supported by a grant of the Swiss National Science Foundation (310030_215606) and an ETH research grant (ETH-17 17-2) to EWB and by ETH core funding of R.G.

## Author contributions

T.v.R., R.Z., and Y.E. contributed equally to this study. T.v.R., R.Z., Y.E., and E.W.B. conceived the project and designed the experiments. T.v.R.

and Y.E. conducted in vitro experiments, protein purification, sample preparation, E.M. grid preparation, and negative stain E.M. R.Z. collected and processed the cryo-EM data, refined the structural models, and finalized figure preparation. P.A. and D.B. assisted in E.M. data collection and provided feedback for E.M. data processing. A.L. conducted the crosslinking mass spectrometry analysis. All authors contributed to the writing and editing of the manuscript.

## Competing interests

The authors declare no competing interests.
