## [Transparent Peer Review file · Nature Communications]

Substrates bind to residues lining the ring of asymmetrically engaged bacterial proteasome activator Bpa

Corresponding Author: Professor Eilika Weber-Ban

Version 0:

Reviewer comments:

Reviewer #1

(Remarks to the Author)

In this paper, the authors use cryo electron microscopy to produce novel, high resolution pictures of the Bpa (also known as PafE) ATP-independent activator of the mycobacteria 20S proteasome core particle. How these types of ATP-independent activators induce degradation is still unknown at a mechanistic level and are likely different from the more well understood ATPase ring complexes that also associate with the 20S core particle. Given the functional similarity of Bpa and the non-ATP dependent activators of the eukaryotic 20S, there is likely shared mechanisms for how these enigmatic factors work. The authors show new structures of the Bpa-20S-substrate complex, Bpa-substrate alone, and generate mutations based on the latter structure that reduce Bpa activity. The major findings are a new asymmetric positioning of Bpa relative to 20S with potentially interesting implications and the positioning of the HspR substrate relative to the Bpa complex. This was a well-described, meticulous study that sheds light on this interesting class of 20S regulators; however, additional verifications of the structural findings with biochemistry seems to be important for the claims made in the manuscript. My major comments are listed below.

Prior to this work, the same authors had solved complexes of Bpa-20S. With the new study, higher resolution snapshots focused on the Bpa/20S interface (with HspR also present) resulted in more details. In particular, more of the C-terminal tail of Bpa known to be important for 20S binding was resolved, suggesting additional interactions between regions of this tail preceding the GQYL motif and the proteasomal H0 helices. This new insight warrants followup investigations with mutations in the H0 helices that should presumably be defective in Bpa binding or mutations along Bpa (162-169).

The other major finding about this interface is the asymmetry of the tethering of the Bpa to the 20S. This 'hinged lid' complex is intriguing and one can envision validating this complex with mixed oligomers of Bpa containing truncated C-terminal tails to show that only 7/12 of the protomers are needed for binding. However, the structural data here is sufficiently compelling that that type of experiment is not critical. But having some test of the importance of this asymmetry would be valuable and gets to the heart of what is the most pressing question in this system which is how binding by Bpa causes degradation of specific substrates by the 20S. For example, does the substrate bind to the tethered side of the Bpa activator when it is being delivered to the 20S or does it bind to the farther side as a 'waiting in queue' type of mechanism? Some insight into the need for this asymmetry for the function of the Bpa mechanism would substantially increase the impact of this work.

The authors should be commended on isolating the Bpa-HspR complex as it seemed to require a substantial amount of optimization and characterization. The mutation studies following from this structural study made it clear that the regions of Bpa implicated in the structure are likely important for HspR binding and degradation of HspR. Unfortunately, it remains a mystery about how binding to Bpa facilitates the specific degradation of this substrate, especially as the structural studies were aimed at answering that question. It is admittedly challenging to see what experiments could be done to resolve this point and so it is understandable that additional work beyond what has been presented here will need to be done to address this.

minor points:

For the Bpa mutants shown in Figure 6, it seems that they are themselves being processed by 20S over time. Is it possible that this autoprocesing is what leads to the reduced overall activity for HspR, especially as it pertains to an inability to completely degrade the substrate? In addition, for Figure 6, a no Bpa control would be important.

Reviewer #2

(Remarks to the Author)

Von Rosen and colleagues aimed with this manuscript to advance our understanding of actinobacterial protein degradation by investigating the cryo-EM structures of the full-length bacterial proteasome activator Bpa from *Mycobacterium tuberculosis* in isolation and asymmetrically bound to the proteasome 20S core particle. Furthermore, cryo-EM structure analysis of the crosslinked complex between a truncated Bpa variant and the natural substrate HspR indicated a potential substrate binding site on the inner surface of the dodecameric Bpa ring that includes residues H131, F138, and R145. Using site-directed mutagenesis and *in vitro* degradation assays, the authors characterized the role of these residues for substrate recruitment to the actinobacterial proteasome.

However, this study provides only limited new insight and is to a large extent confirmatory of previous findings. In addition, there are weaknesses in the experimental data for several of the conclusions. Overall, this manuscript seems therefore not suited for publication in *Nature Communications*.

Major points:

1) The authors claim that 7 of the 12 Bpa subunits simultaneously contact all 7 binding pockets of the CP alpha ring, yet there is no solid experimental evidence for that. In the non-symmetrized reconstruction of the crosslinked Bpa-20S CP complex (Fig. 1D, F), the densities for Bpa's C-terminal linkers and tails are over-sharpened, do not resemble clear protein structure, and likely originate from averaging various different tail conformations and contacts. The concentric appearance of this connecting density between Bpa and 20S CP, with the narrowest part close to the Bpa ring, also seems inconsistent with 7 neighboring Bpa subunits simultaneously docking onto the 20S CP.

These doubts about fully occupied binding pockets of the CP alpha ring are further supported by the higher-resolution structure with imposed D7 symmetry (Fig. 2), where the densities for Bpa's C-terminal tails and GQYL motifs are much weaker than the surrounding 20S CP density, most likely due to the averaging of occupied and unoccupied binding pockets. Overall, there is only an incremental advance compared to previous cryo-EM structures of a truncated Bpa/PafE bound to 20S CP and full-length PafE in complex with an open gate 20S CP mutant, which showed the CP pockets occupied with PafE's GQYL motifs.

2) Similarly, there is not much new to learn from the cryo-EM structure of the isolated Bpa. The authors emphasize that their cryo-EM reconstruction was done with full-length Bpa, whereas previous X-ray crystallography studies used truncated Bpa variants (missing 14 N-terminal and 21 C-terminal residues). However, in the presented new structure, the 35 N-terminal residues and 33 C-terminal residues are disordered and not resolved anyway, such that this reconstruction is largely consistent with the existing crystal structure. As an important difference to the previous structure, the authors highlight that the last 8 residues of helix H4's C-terminal helical extension are highly flexible, implying some functional relevance of this difference. Interestingly, the authors later use the crystal structure and the existence of the extended H4 helix as an argument for the potential involvement of Bpa's R145 in substrate binding, as this residue "is situated two helical turns below F138...", yet was not present in the truncated Bpa variant used for crosslinking to the HspR substrate.

3) To identify the substrate binding site on Bpa, the authors performed cryo-EM analyses of the glutaraldehyde crosslinked complex between truncated Bpa and HspR. They observed a small additional density with low resolution and high heterogeneity on the inner surface of the Bpa ring, in an area close to H131. This density is also in proximity to F138, a residue that was already previously identified in functional studies to be important for substrate binding, yet was not resolved in the reconstruction presented here. Based on the existing crystal structure with an extended H4 helix, the authors postulated that in addition to H131 and F138, residue R145 may be involved in HspR binding to Bpa.

The authors tested the role of these 3 residues by generating the corresponding single, double, and triple mutants of Bpa, and characterizing HspR degradation by the Bpa-20S CP complexes. Although these mutants showed reduced HspR degradation activity (Fig. 6), it remains unclear to what extent this is indeed caused by the disruption of specific substrate-binding interactions. The authors state that CD measurement confirmed the alpha-helical structure of the Bpa mutants, yet it is ignored that the normalized ellipticity is less negative for all variants and especially for the double and triple mutants (by almost 25%, Fig. S3). It is therefore possible that the mutations disrupt the H4 helix, rather than prevent residue-specific ring stacking or electrostatic interactions with HspR. Furthermore, the concentration of the triple mutant Bpa used in the HspR degradation assay is clearly much lower (< 50%) than that of wild type or other Bpa mutants (Figure 6A).

To measure the HspR affinity of the Bpa mutants, the authors used fluorescently labeled Bpa in spectral shift binding assays. Although the K_d determined for wild type Bpa is similar to a value previously reported by the same group, it is not considered that the lysine-specific fluorescence labeling of Bpa with the bulky RED dye may interfere with HspR binding. The authors found in their BS3 crosslinking and cryo-EM structural studies that Lys55 is in direct proximity to HspR, and modification with RED NHS in this position may therefore affect the K_d for HspR binding.

Overall, the presented studies extend our knowledge about the HspR binding site on Bpa by indicating that in addition to the previously identified F138 residue, the neighboring H131 and R145 in the H4 helix may also be involved in substrate interactions. However, this appears to be a rather incremental advance.

Reviewer #3

(Remarks to the Author)

The manuscript describes the structure of the bacterial proteasome activator Bpa from *M. Tuberculosis*. The authors record

the structure using a 3D-cryo-EM reconstruction of the Bpa and its natural substrate HspR at a resolution of 4.1 Å. Additionally, the authors use cross-linking mass spectrometry to determine if the additional mass that is visible in the EM structure is indeed the HspR substrate.

Overall the manuscript is well written and the authors are able to show nicely the complex between the proteasome, the activator and its substrate. All figures are of high quality and support the claims laid out in the paper. I recommend publishing the manuscript.

Minor changes

The section describing the material and methods for the mass spectrometry x-linking experiments is very short and needs to be extended so the experiment can be repeated in other laboratories.

Version 1:

Reviewer comments:

Reviewer #1

(Remarks to the Author)

The authors have addressed my major concerns regarding the Bpa binding to the 20S core. While it still remains unclear how the nature of the asymmetric interaction of Bpa/20S affects the ability to degrade HspR, it is appreciated that this is beyond the scope of this paper. Overall, I would encourage acceptance of this paper.

Reviewer #2

(Remarks to the Author)

In their rebuttal and revised manuscript the authors well addressed the criticisms, concerns, and questions raised in my review and the review of referee #1. I still think that the advance and extend of concrete mechanistic insight is rather slim for a Nature Communications manuscript, and the authors themselves state that their study provides a promising basis for future mechanistic work.

However, given the strong support by referees 1 and 3, I am also approving further consideration or potential acceptance for publication in Nature Communications.

Reviewer #3

(Remarks to the Author)

The authors answered all my comments. I recommend to publish the paper in its current form.

REVIEWER COMMENTS – POINT-BY-POINT REPLIES

We thank the reviewers for their time in carefully evaluating the manuscript and for their constructive comments.

Reviewer #1 (Remarks to the Author):

In this paper, the authors use cryo electron microscopy to produce novel, high resolution pictures of the Bpa (also known as PafE) ATP-independent activator of the mycobacteria 20S proteasome core particle. How these types of ATP-independent activators induce degradation is still unknown at a mechanistic level and are likely different from the more well understood ATPase ring complexes that also associate with the 20S core particle. Given the functional similarity of Bpa and the non-ATP dependent activators of the eukaryotic 20S, there is likely shared mechanisms for how these enigmatic factors work. The authors show new structures of the Bpa-20S-substrate complex, Bpa-substrate alone, and generate mutations based on the latter structure that reduce Bpa activity. The major findings are a new asymmetric positioning of Bpa relative to 20S with potentially interesting implications and the positioning of the HspR substrate relative to the Bpa complex. This was a well-described, meticulous study that sheds light on this interesting class of 20S regulators; however, additional verifications of the structural findings with biochemistry seems to be important for the claims made in the manuscript. My major comments are listed below.

Prior to this work, the same authors had solved complexes of Bpa-20S. With the new study, higher resolution snapshots focused on the Bpa/20S interface (with HspR also present) resulted in more details. In particular, more of the C-terminal tail of Bpa known to be important for 20S binding was resolved, suggesting additional interactions between regions of this tail preceding the GQYL motif and the proteasomal H0 helices. This new insight warrants followup investigations with mutations in the H0 helices that should presumably be defective in Bpa binding or mutations along Bpa (162-169).

The primary contribution to the binding of Bpa to the 20S CP comes from the insertion of the Bpa GQYL motifs into the binding pockets located between the proteasome α subunits. Truncation of the C-terminal tails by the GQYL motif abolishes interaction at

physiological concentrations (micromolar range). However, to address whether additional contacts contribute to the interaction, we mutated residues on the H0 helix that are positioned towards the Bpa C-terminal tails (E10A-R14A). Using the GraFix protocol, we then generated a complex of the 20S proteasome containing either the mutated or wild-type α -subunits with Bpa and analyzed the samples using negative stain EM. The 2D classes from these samples showed that 81.3% of proteasomes containing wild-type α -subunits had Bpa bound at both ends, while this was the case for only 31% of the proteasomes containing the H0 α -subunit variant. This experiment supports our hypothesis that the H0 helix is involved in the interaction between the proteasome and Bpa. The data from this experiment have been added to the supplement of the manuscript as part of Figure S3.

Bpa WT binding to 20S CP

Bpa WT binding to E10A-R14A 20S CP

The other major finding about this interface is the asymmetry of the tethering of the Bpa to the 20S. This 'hinged lid' complex is intriguing and one can envision validating this complex with mixed oligomers of Bpa containing truncated C-terminal tails to show that only 7/12 of the protomers are needed for binding. However, the structural data here is sufficiently compelling that that type of experiment is not critical. But having some test of the importance of this asymmetry would be valuable and gets to the heart of what is the most pressing question in this system which is how binding by Bpa causes degradation of specific substrates by the 20S.

The symmetry mismatch between Bpa and the 20S CP in the complex is 12:7, meaning that no more than seven of the twelve Bpa tails can bind, before the α -ring is fully occupied. Our observation of an axially offset Bpa ring, with seven consecutive tails occupying all seven binding pockets of the α -ring, highlights not so much the number of tails required but their distribution in Bpa.

As suggested by the reviewer, we generated a construct employed in a previous study (Hu et al. (JBC, 2018, DOI: 10.1074/jbc.RA117.001471)), in which C-terminally truncated Bpa (Bpa1-155) is covalently linked to full-length Bpa (Bpa1-174) through a linker of five amino acids in length. The covalent dimer forms the 12-meric Bpa ring like wild-type Bpa, but instead of the twelve C-terminal tails bearing the GQYL-interaction motif, it features only six evenly distributed C-terminal tails. In the previous study, the authors demonstrated that while this variant forms rings, it does not support the degradation of HspR in the presence of 20S proteasome. However, they did not test whether this defect was due to an inability to form a complex with 20S CP. It is expected that the lower avidity, due to the reduced total number of tails, reduces the affinity of Bpa for the 20S CP. The fact that the six tails are not consecutive may also play a role, though it would be challenging to separate this effect from that of avidity. However, we might observe a difference in the shift of Bpa from the central axis. To test this, we generated a complex of inactive full-length 20S CP with the engineered Bpa variant, using the GraFix protocol, and compared it to the complex with wild-type Bpa using negative stain EM (see figure below). Our results show that the Bpa Δ C-Bpa ring has a substantially lower affinity for the CP as nearly 73% of proteasomes are uncapped and the 27% of proteasomes that associate with the Bpa Δ C-Bpa ring only do so on one side (singly capped). In contrast, in the Bpa wild-type dataset, less than 1% of proteasomes are uncapped, while about 80% of proteasomes are doubly capped by Bpa rings and 18% singly capped. We have added this experiment to the supplement of the manuscript in Figure S3.

Bpa WT binding to 20S CP

Bpa Δ C-Bpa binding to 20S CP

Interestingly, the 3D reconstruction (lower panel) reveals that the proteasome complexes formed with the mixed oligomer still exhibit asymmetric tethering and a hinged lid conformation. However, not all six available tails are involved in the interaction, as indicated by the narrower density in the EM map between the Bpa ring and the CP (panel C, black arrow). This could suggest that also in the case of the mixed oligomer only the tails from one side of the Bpa ring are engaged.

For example, does the substrate bind to the tethered side of the Bpa activator when it is being delivered to the 20S or does it bind to the farther side as a 'waiting in queue' type of mechanism? Some insight into the need for this asymmetry for the function of the Bpa mechanism would substantially increase the impact of this work.

These are compelling questions regarding the molecular mechanisms by which Bpa processes its substrate. The current manuscript presents intriguing possibilities and lays a foundation for future investigations. However, addressing these questions falls beyond the scope of this work.

The authors should be commended on isolating the Bpa-HspR complex as it seemed to require a substantial amount of optimization and characterization. The mutation studies following from this structural study made it clear that the regions of Bpa implicated in the structure are likely important for HspR binding and degradation of HspR. Unfortunately, it remains a mystery about how binding to Bpa facilitates the specific degradation of this substrate, especially as the structural studies were aimed at answering that question. It is admittedly challenging to see what experiments could be done to resolve this point and so it is understandable that additional work beyond what has been presented here will need to be done to address this.

We fully agree with the reviewer that future mechanistic studies will be needed to get a complete picture about the substrate processing by this ATP-independent degradation machine, requiring dynamic information in addition to the structural analysis presented here.

Minor points:

For the Bpa mutants shown in Figure 6, it seems that they are themselves being processed by 20S over time. Is it possible that this autoprocesing is what leads to the reduced overall activity for HspR, especially as it pertains to an inability to completely degrade the substrate? In addition, for Figure 6, a no Bpa control would be important.

While we do not observe autoprocesing of Bpa by the proteasome, we agree with the reviewer that the amount of Bpa in one of the panels appears slightly lower than in the others. To address this, we repeated the entire set of degradation experiments again in triplicates and included the requested control experiment where Bpa was omitted. Representative gels and densitometric analysis of HspR band intensity can be found in the updated version of Figure 6 (also depicted below). Our data confirm that HspR is more stable against proteasomal degradation in the absence of Bpa than in its presence, consistent with our previous observations reported in von Rosen et al., 2023 (LSA, DOI: 10.26508/lsa.202301923).

A**B****C**
Reviewer #2 (Remarks to the Author):

Von Rosen and colleagues aimed with this manuscript to advance our understanding of actinobacterial protein degradation by investigating the cryo-EM structures of the full-length bacterial proteasome activator Bpa from Mycobacterium tuberculosis in isolation and asymmetrically bound to the proteasome 20S core particle. Furthermore, cryo-EM structure analysis of the crosslinked complex between a truncated Bpa variant and the natural substrate HspR indicated a potential substrate binding site on the inner surface of the dodecameric Bpa ring that includes residues H131, F138, and R145. Using site-directed mutagenesis and in vitro degradation assays, the authors characterized the role of these residues for substrate recruitment to the actinobacterial proteasome.

However, this study provides only limited new insight and is to a large extent confirmatory of previous findings. In addition, there are weaknesses in the experimental data for several of the conclusions. Overall, this manuscript seems therefore not suited for publication in Nature Communications.

Major points:

1) The authors claim that 7 of the 12 Bpa subunits simultaneously contact all 7 binding pockets of the CP alpha ring, yet there is no solid experimental evidence for that. In the non-symmetrized reconstruction of the crosslinked Bpa-20S CP complex (Fig. 1D, F), the densities for Bpa's C-terminal linkers and tails are over-sharpened, do not resemble clear protein structure, and likely originate from averaging various different tail conformations and contacts. The concentric appearance of this connecting density between Bpa and 20S CP, with the narrowest part close to the Bpa ring, also seems inconsistent with 7 neighboring Bpa subunits simultaneously docking onto the 20S CP. These doubts about fully occupied binding pockets of the CP alpha ring are further supported by the higher-resolution structure with imposed D7 symmetry (Fig. 2), where the densities for Bpa's C-terminal tails and GQYL motifs are much weaker than the surrounding 20S CP density, most likely due to the averaging of occupied and unoccupied binding pockets.

Overall, there is only an incremental advance compared to previous cryo-EM structures of a truncated Bpa/PafE bound to 20S CP and full-length PafE in complex

with an open gate 20S CP mutant, which showed the CP pockets occupied with PAF_E's GQYL motifs.

Due to differences in local resolution between the Bpa and CP regions in the Bpa-20S CP map, it is challenging to depict the Bpa-CP interface. For illustrative purposes, we used the OccuPy software to enhance the Bpa density relative to CP density and to exclude solvent noise, but no sharpening was applied. We agree with the reviewer that the noisy densities observed for the Bpa tails likely stem from some level of conformational heterogeneity in the region beyond the GQYL motif, and we added a comment in the text to address this.

Nevertheless, our structure provides reliable and meaningful insights into the relative positioning of Bpa on the proteasomal α -rings and the topology of the Bpa C-terminal densities. Specifically, we observe asymmetric positioning of Bpa with respect to the central axis of the CP α -ring, along with clustering of the Bpa tail densities on one side of Bpa. This results in positioning of the Bpa side providing the seven C-terminal tails for binding over the α -ring pore, which leads to the concentric appearance of the connecting density. This observation is further supported by our experiment using negatively stained grid preparations, where, despite the lower resolution of the 3D image reconstructions, the asymmetric positioning and the connectivity of the tails to one side of the Bpa ring are clearly visible. To better illustrate this, we have changed panel F in Figure 1 to show a top view of the offset.

Our cryo-EM analysis of the Bpa-20S CP complex (without applying 7-fold symmetry) indicates that all binding pockets of the CP α -ring are occupied by the GQYL motif of Bpa. The weaker signal observed for these motifs compared to the CP can be attributed to a degree of conformational heterogeneity in the linkers connecting the GQYL motif with helix H4 of Bpa. In response to the reviewer's comments regarding the connecting densities between Bpa and the 20S CP, we performed negative stain EM analyses of complexes containing H0 α -ring mutations and with a mixed Bpa Δ C-Bpa ring (see answer to reviewer 1 and supplementary Figure S3 in the manuscript).

The asymmetric positioning and nature of the Bpa-CP complex suggests intriguing mechanistic implications, potentially indicating a wobbling-rolling motion of Bpa during substrate processing. Our results therefore provide a promising basis for future mechanistic studies, which are however beyond the scope of the current work.

2) Similarly, there is not much new to learn from the cryo-EM structure of the isolated Bpa. The authors emphasize that their cryo-EM reconstruction was done with full-length Bpa, whereas previous X-ray crystallography studies used truncated Bpa variants (missing 14 N-terminal and 21 C-terminal residues). However, in the presented new structure, the 35 N-terminal residues and 33 C-terminal residues are disordered and not resolved anyway, such that this reconstruction is largely consistent with the existing crystal structure. As an important difference to the previous structure, the authors highlight that the last 8 residues of helix H4's C-terminal helical extension are highly flexible, implying some functional relevance of this difference. Interestingly, the authors later use the crystal structure and the existence of the extended H4 helix as an argument for the potential involvement of Bpa's R145 in substrate binding, as this residues "is situated two helical turns below F138...", yet was not present in the truncated Bpa variant used for crosslinking to the HspR substrate.

Our 3D reconstruction represents the first high-resolution cryo-EM map (3.2 Å global resolution) of the Bpa dodecamer. Although the primary focus of our study was to characterize the Bpa-HspR complex, solving the structure of full-length Bpa was an important step to address potential structural features not previously observable for the N-terminal and C-terminal extensions in the crystal structures, and to thereby validate the use of truncation variants.

While it is true that the N-terminal 35 residues and C-terminal 33 residues are disordered in our structure, this finding shows that the C-terminal extensions are highly dynamic. This dynamic behavior could not be inferred from the previous truncated Bpa variants used in X-ray crystallography. The fact that the helix 4 extension forms a longer helical stretch in the crystal structure suggests it exists as a conformational ensemble with helix propensity, and that it can fluctuate between a helical conformation and more disordered conformations. We do not see such dynamic behavior as a contradiction.

In light of the reviewer's comment, we have revised the text to clarify that our intent was not to imply a discrepancy with the existing crystal structures. Instead, our cryo-EM structure complements previous data by showing that the C-terminal extensions are indeed flexible and capable of existing in multiple conformations.

3) To identify the substrate binding site on Bpa, the authors performed cryo-EM analyses of the glutaraldehyde crosslinked complex between truncated Bpa and HspR. They observed a small additional density with low resolution and high heterogeneity on the inner surface of the Bpa ring, in an area close to H131. This density is also in proximity to F138, a residue that was already previously identified in functional studies to be important for substrate binding yet was not resolved in the reconstruction presented here. Based on the existing crystal structure with an extended H4 helix, the authors postulated that in addition to H131 and F138, residue R145 may be involved in HspR binding to Bpa.

The authors tested the role of these 3 residues by generating the corresponding single, double, and triple mutants of Bpa, and characterizing HspR degradation by the Bpa-20S CP complexes. Although these mutants showed reduced HspR degradation activity (Fig. 6), it remains unclear to what extent this is indeed caused by the disruption of specific substrate-binding interactions. The authors state that CD measurement confirmed the alpha-helical structure of the Bpa mutants, yet it is ignored that the normalized ellipticity is less negative for all variants and especially for the double and triple mutants (by almost 25%, Fig. S3). It is therefore possible that the mutations disrupt the H4 helix, rather than prevent residue-specific ring stacking or electrostatic interactions with HspR. Furthermore, the concentration of the triple mutant Bpa used in the HspR degradation assay is clearly much lower (< 50%) than that of wild-type or other Bpa mutants (Figure 6A).

We appreciate the reviewer's observations and would like to clarify several points. As mentioned in our manuscript, residue F138 has been previously identified as important for the degradation of HspR by the Bpa-CP complex. It is important to note that the earlier study did not investigate the binding of HspR to Bpa. The fact that F138 has been implicated in related contexts across different studies reinforces its functional significance and should be considered a point of affirmation rather than criticism.

Regarding the CD spectroscopy data, we agree that Bpa's low molar extinction coefficient makes the method particularly sensitive to variations in protein concentration. To address the reviewer's concern about potential discrepancies in the CD measurements, we repeated the experiments with careful attention to ensuring accurate protein concentration across all variants and replicates. As a result, Supplementary Figure 3 has been updated with this new data (see panel below) and

is now shown as supplementary Figure S4. The differences in ellipticity between the wild-type Bpa and its variants, though observable, are minor and fall within the inherent error margin of the method.

To measure the HspR affinity of the Bpa mutants, the authors used fluorescently labeled Bpa in spectral shift binding assays. Although the K_d determined for wild-type Bpa is similar to a value previously reported by the same group, it is not considered that the lysine-specific fluorescence labeling of Bpa with the bulky RED dye may interfere with HspR binding. The authors found in their BS3 crosslinking and cryo-EM structural studies that Lys55 is in direct proximity to HspR, and modification with RED NHS in this position may therefore affect the K_d for HspR binding.

We understand the reviewer's concern regarding the potential interference of the fluorescent dye with HspR binding. However, as the reviewer acknowledges, we observe binding for wild-type HspR to Bpa measured by the same method, which serves as a positive control. The observed decrease in binding affinity for the Bpa variants is therefore due to the introduced mutations rather than any interference from the fluorescent dye.

Overall, the presented studies extend our knowledge about the HspR binding site on Bpa by indicating that in addition to the previously identified F138 residue, the neighboring H131 and R145 in the H4 helix may also be involved in substrate interactions. However, this appears to be a rather incremental advance.

Our manuscript provides a structural analysis of HspR-bound Bpa by cryo-electron microscopy, offering insights into the interaction of Bpa with HspR prior to its handover

into the 20S CP. The structural data are combined with biochemical experiments and crosslinking mass spectrometry.

Furthermore, our cryo-EM analysis of the Bpa-CP/HspR complex reveals asymmetric positioning of Bpa relative to the rotational symmetry axis of the α -ring, as well as a hinged arrangement of Bpa. These findings have intriguing mechanistic implications that extend beyond the current understanding and form a solid basis for future studies into Bpa-mediated substrate processing. Given these significant contributions, we believe our work represents more than an incremental advance in the field.

Reviewer #3 (Remarks to the Author):

The manuscript describes the structure of the bacterial proteasome activator Bpa from *M. Tuberculosis*. The authors record the structure using a 3D-cryo-EM reconstruction of the Bpa and its natural substrate HspR at a resolution of 4.1 Å. Additionally, the authors use cross-linking mass spectrometry to determine if the additional mass that is visible in the EM structure is indeed the HspR substrate.

Overall the manuscript is well written and the authors are able to show nicely the complex between the proteasome, the activator and its substrate. All figures are of high quality and support the claims laid out in the paper. I recommend publishing the manuscript.

Minor changes:

The section describing the material and methods for the mass spectrometry x-linking experiments is very short and needs to be extended so the experiment can be repeated in other laboratories.

We agree with the reviewer that this part of the method section should have been more elaborate. We have expanded the section and now describe the crosslinking mass spectrometry experiments in more detail.